# Trajectory Inference with Smooth Schrödinger Bridges

**Wanli Hong** [* 1 2]  **Yuliang Shi** [* 3]  **Jonathan Niles-Weed** [1 4]

## Abstract

Motivated by applications in trajectory inference and particle tracking, we introduce *Smooth Schrödinger Bridges*. Our proposal generalizes prior work by allowing the reference process in the multi-marginal Schrödinger Bridge problem to be a smooth Gaussian process, leading to more regular and interpretable trajectories in applications. Though naïvely smoothing the reference process leads to a computationally intractable problem, we identify a class of processes (including the Matérn processes) for which the resulting Smooth Schrödinger Bridge problem can be *lifted* to a simpler problem on phase space, which can be solved in polynomial time. We develop a practical approximation of this algorithm that outperforms existing methods on numerous simulated and real single-cell RNAseq datasets.

## 1. Introduction

The *trajectory inference problem*—reconstructing the paths of particles from snapshots of their evolution—is fundamental to modern data science, with applications in single-cell biology (Huguet et al., 2022a; Saelens et al., 2019; Schiebinger et al., 2019; Sha et al., 2023; Tong et al., 2020), fluid dynamics (Brunton et al., 2020; Ouellette et al., 2006), and astronomical object tracking (Kubica et al., 2007; Liounis et al., 2020). Given observations of a collection of indistinguishable particles at discrete times, the statistician aims to infer the continuous trajectories that generated this data.

A leading approach (Chen et al., 2019; Chizat et al., 2022;

---

[*]Equal contribution [1]Center for Data Science, New York University, New York, United States [2]Shanghai Frontiers Science Center of Artificial Intelligence and Deep Learning New York University Shanghai Shanghai, China [3]Department of Mathematics, The University of British Columbia, Vancouver, Canada [4]Courant Institute of Mathematical Science, New York University, New York, United States. Correspondence to: Jonathan Niles-Weed <jnw@cims.nyu.edu>.

*Proceedings of the $42^{nd}$ International Conference on Machine Learning*, Vancouver, Canada. PMLR 267, 2025. Copyright 2025 by the author(s).

Lavenant et al., 2024) builds on Schrödinger's remarkable thought experiment (Schrödinger, 1931; Schrödinger, 1932), formulating trajectory inference as a Kullback–Leibler minimization problem. Suppose that a set of particles is observed at times $t_0, \ldots, t_K \in [0, 1]$ and that the particles' arrangement at time $t_k$ is represented by a probability measure $\mu_k$, for all $k \in [K]$. The *multi-marginal Schrödinger bridge* corresponding to these observations is the solution of

$$\min_{P \in \mathcal{P}(\Omega)} \mathrm{D}(P|R) \text{ s.t. } P_k = \mu_k, \ \forall k \in [K], \qquad (1)$$

where $D$ is the Kullback–Leibler divergence, $P_k$ denotes the time marginal of $P$ at time $t_k$, and $R$ is the law of a reference process, typically Brownian motion. (See Appendix A for a complete list of notation.) This formulation has an elegant quasi-Bayesian interpretation: the Schrödinger bridge matches the observed marginal distributions while keeping particle trajectories as faithful as possible to the prior $R$.

While extensively studied both theoretically and methodologically, this classical Schrödinger Bridge (SB) approach has a critical limitation: its trajectories inherit the roughness of Brownian paths, leading to noisier estimators and less interpretable posterior paths; see Figure 1, left side. Moreover, when the experimenter aims to track the positions of individual particles in a system evolving over time, inference with the Brownian motion prior fails to "borrow strength" from adjacent time points, resulting in less accurate results; see Figure 2, left side.

The literature on trajectory inference contains various proposals for encouraging smooth paths, inspired by spline algorithms on $\mathbb{R}^d$ or the dynamics of physical systems. (A full comparison with prior work appears in Appendix B.) However, from the perspective of Schrödinger's original formulation, these modifications sacrifice the clear statistical interpretation of (1). We develop a new approach: a smooth version of the Schrödinger Bridge problem in which the Brownian motion is replaced by a smooth Gaussian process. This proposal has an appealing statistical grounding: like Gaussian process regression (Rasmussen & Williams, 2006), it provides a flexible and principled way to perform non-parametric estimation while incorporating prior smoothness assumptions. As Figures 1 and 2, right side, show, the resulting estimates significantly outperform the vanilla Schrödinger Bridge, producing better estimates with smoother paths.

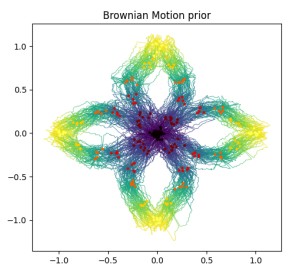
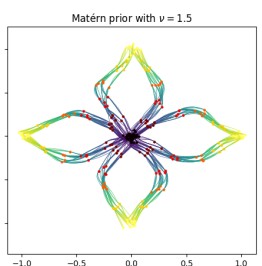

*Figure 1.* Comparison between the classical SB and smooth SB with Matérn prior on a trajectory inference task for the petal dataset (Huguet et al., 2022a). Colored points are observed data and paths are inferred trajectories. Our smooth SB approach generates much smoother and more concentrated trajectories.

**Our contributions:**

- We propose a class of Gaussian processes suitable as priors for smooth SB problems, which we show can be *lifted* to simpler SB problems on phase space.

- We show that a message passing algorithm for the resulting lifted SB problem converges in time quadratic in $K$, an exponential improvement over the baseline approach.

- We develop an efficient approximation of our algorithm that outperforms existing methods in practice, in several cases by 2–5x.

## 2. Background

### 2.1. The Multi-Marginal Schrödinger Bridge

Let $R$ be a measure on the space $\Omega = C([0,1];\mathbb{R}^d)$ of continuous $\mathbb{R}^d$-valued paths, and let $\mu_0, \ldots, \mu_K$ be probability measures on $\mathbb{R}^d$. Fix a sequence of times $t_0 = 0 < t_1 \cdots < t_K = 1$. Given $\omega \in \Omega$, we write $\omega_k = \omega(t_k)$ for $k \in [K]$. Given a probability measure $P$ on $\Omega$ and $k \in [K]$, let $P_k$ be the marginal distribution of $P$ at time $t_k$, that is, let $P_k$ be the element of $\mathcal{P}(\mathbb{R}^d)$ obtained by pushing $P$ forward under the map $\omega \mapsto \omega_k$. The multi-marginal Schrödinger Bridge (1) exists under suitable moment conditions on $\mu_0, \ldots, \mu_K$ (Léonard, 2014), and the strict convexity of the Kullback–Leibler divergence guarantees that when a solution exists, it is unique.

Though phrased as a minimization problem over the space $\mathcal{P}(\Omega)$, the Schrödinger bridge problem admits a "static" reformulation as a multi-marginal entropic optimal transport problem over the space $\mathcal{P}(\mathbb{R}^{d(K+1)})$. Write $R_{[K]}$ for the joint law of $\boldsymbol{\omega} := (\omega_0, \ldots, \omega_K)$ for $\omega \sim R$.

**Lemma 2.1.** *There is an one-to-one correspondence be-*

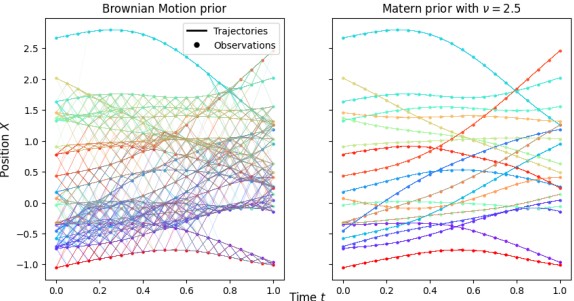

*Figure 2.* Comparison between the classical SB and smooth SB. The observations come from 20 independent trajectories of a Matérn Gaussian process with $\nu = 3.5$. The two pictures depict the posterior identification of particles obtained by solving the SB problem with two different priors. Our smooth SB approach recovers trajectories much more accurately.

*tween solutions to* (1) *and solutions to*

$$\min_{P_{[K]} \in \mathcal{P}(\mathbb{R}^{d(K+1)})} \mathrm{D}(P_{[K]}|R_{[K]}), \ P_k = \mu_k, \ \forall k \in [K]. \quad (2)$$

*Moreover, if each measure $\mu_k$ is absolutely continuous with finite entropy and $R_{[K]}$ has density $\exp(-C(\boldsymbol{\omega}))$, then this problem is equivalent to*

$$\min_{P_{[K]} \in \mathcal{P}(\mathbb{R}^{d(K+1)})} \int C(\boldsymbol{\omega})P_{[K]}(\boldsymbol{\omega}) \, d\boldsymbol{\omega} + \mathrm{D}\left(P_{[K]} \Big| \bigotimes_{k=0}^{K} \mu_k\right)$$
$$P_k = \mu_k, \ \forall k \in [K]. \quad (3)$$

The reformulation in Lemma 2.1 is essential because it eliminates the need to optimize over probability measures on the infinite-dimensional space $\Omega$. Though (3) was derived for absolutely continuous measures, the expression is sensible for arbitrary marginal measures. Following a common convention in the literature (see, e.g., Pooladian & Niles-Weed, 2024), we therefore take (3) as the basic definition of the multi-marginal Schrödinger bridge in what follows. Importantly, when the measures $\mu_k$ are finitely supported, the resulting optimization problem is finite-dimensional.

### 2.2. Sinkhorn's algorithm for multi-marginal transport

Specializing to the case where each of the marginals $\mu_k$ is supported on a finite set $\mathcal{X}_k$ of size at most $n$, the joint measure $P_{[K]}$ can be represented by a finite—albeit exponentially large—order $(K+1)$ tensor, of total size $n^{K+1}$.

A direct method to solve (3) in this case uses Sinkhorn's celebrated scaling Algorithm 1. Sinkhorn's algorithm is based on the observation that the optimal solution $P_{[T]}^*$ to (3) is of the form $P_{[T]}^*(\mathbf{x}) = \exp(-C(\mathbf{x}))\prod_{k=0}^{K} v_k^*(x_k)$ for all $\mathbf{x} = (x_0, \ldots, x_K) \in \prod_{k=0}^{K} \mathcal{X}_k$, for some "scaling functions" $v_k^* : \mathcal{X}_k \to \mathbb{R}^+, k \in [K]$. Algorithm 1 produces a

sequence of iterates $v_k^{(\ell)}$, $k \in [K]$, $\ell \geq 0$, which converge to these optimal scalings.

---

**Algorithm 1** Vanilla Sinkhorn Algorithm

---

**Input:** $\mu_0, ..., \mu_K$
Initialize $v_0^{(0)}, ..., v_K^{(0)} \in (\mathbb{R}^+)^n$, $\ell = 0$.
**while** not converged **do**
   $\ell \leftarrow \ell + 1$
   **for** $k = 0, ..., K$ **do**
      Compute $\mathbf{S}_k^{(\ell)}$ using (4) with $\left\{v_i^{(\ell)}\right\}_{i<k}$ and
      $\left\{v_i^{(\ell-1)}\right\}_{i>k}$
      $v_k^{(\ell)} \leftarrow \mu_k \oslash \mathbf{S}_k^{(\ell)}$
   **end for**
**end while**
Return $(v_k^{(\ell)})_{k \in [K]}$

---

The key subroutine in Algorithm 1 is an operation called "marginalization": given current iterates $(v_i)_{i \in [K]}$, the marginalization along coordinate $k$ is the function $\mathbf{S}_k : \mathcal{X}_k \to \mathbb{R}^+$ defined by

$$\mathbf{S}_k(x_k) := \sum_{\substack{i=0 \\ i \neq k}}^{K} \sum_{x_i \in \mathcal{X}_i} \exp(-C(\mathbf{x})) \prod_{\substack{i=0 \\ i \neq k}}^{K} v_i(x_i). \quad (4)$$

The complexity of computing $\mathbf{S}_k$ directly is exponential in $K$, since the sum in (4) has an exponential number of terms. Moreover, no *general* polynomial-time algorithms for even approximating (4) exist under standard computational complexity assumptions (Altschuler & Boix-Adserà, 2023, Lemma 3.7). Together, these facts suggest that (3) will only be computationally feasible under special assumptions on $R$.

A crucial observation (Altschuler & Boix-Adserà, 2023; Chizat et al., 2022), which has driven the near-universal choice of Brownian motion as a prior, is that when $R$ is a *Markov* process, (4) can be computed in $\mathrm{Poly}(K, |\mathcal{X}|)$ time. The tractability of marginalization when $R$ is a Markov process follows from the decomposition

$$\exp(-C(\mathbf{x})) = R_{[K]}(\mathbf{x}) = R_0(x_0) \prod_{k=1}^{K} R_{k|k-1}(x_k|x_{k-1}),$$

which shows that the exponential-size sum in (4) factors into $K$ sums, each of which can be computed efficiently. The assumption that $R$ is Markov extends even to works that consider priors other than Brownian motion (Bunne et al., 2023; Vargas et al., 2021).

This raises what appears to be an inherent tension: efficient algorithms for (3) rely on the assumption that the process $R$ is Markov; however, limiting to the case of Markov processes necessarily precludes consideration of priors $R$ with

smooth sample paths (see Lemma C.1). This observation suggests the pessimistic conclusion that *no* practical algorithm exists for solving Schrödinger bridges with smooth priors.

However, the key contribution of this work is to show that this conclusion is *false*. The next section identifies a class of smooth priors for which (3) can be solved efficiently.

### 2.3. Autoregressive Gaussian processes

To develop our proposal for efficient smooth Schrödinger bridges, we focus on the class of *continuous-time Gaussian autoregressive processes*, a classic model in statistics and signal processing (Phillips, 1959). They are also a widespread choice in applications: as we show below (Theorem 2.3 and Proposition 2.4), the famous *Matérn kernel* gives rise to such processes. While it is known that such processes offer crucial efficiency benefits in Gaussian process regression (Gilboa et al., 2015), their algorithmic implications for the SB problem have not been explored prior to this work.

**Definition 2.2** (Rasmussen & Williams, 2006, Appendix B)**.** Let $m$ be a positive integer. A Gaussian process $t \mapsto \omega(t)$ defined on $C^{m-1}(\mathbb{R}; \mathbb{R}^d)$ is a *Gaussian autoregressive process* (GAP) of order $m$ if it is a stationary solution to the stochastic differential equation

$$\frac{d^m}{dt^m}\omega(t) + a_{m-1}\frac{d^{m-1}}{dt^{m-1}}\omega(t) + \cdots + a_0\omega(t) = \sigma\xi(t),$$

for some constants $a_0, \ldots, a_{m-1} \in \mathbb{R}$ and $\sigma \in \mathbb{R}^{d \times d}$, where $\xi(t)$ denotes a white-noise process on $\mathbb{R}^d$ with independent coordinates.

The key fact about GAPs, which we leverage to develop efficient algorithms, is that, even though a GAP $\omega$ is typically not Markov, the process $\eta := (\omega, d\omega/dt, \ldots, d^{m-1}\omega/dt^{m-1})$ taking values in phase space $\mathbb{R}^{d \times m}$ *is* a Markov process. We call $\eta$ the "lifted" Gaussian process corresponding to $\omega$. The main observation driving our practical algorithm for smooth Schrödinger bridges is that when $R$ is a GAP, problem (3) admits an efficient algorithm obtained by *lifting* the optimization problem to phase space.

GAPs form a rich class of smooth Gaussian processes. Moreover, they admit a simple characterization in terms of their covariance functions. Recall that a Gaussian process $\omega$ taking values in $C(\mathbb{R}; \mathbb{R}^d)$ is characterized by two functions $m : \mathbb{R} \to \mathbb{R}^d$ and $\kappa : \mathbb{R}^2 \to \mathbb{R}^{d \times d}$, which satisfy

$$m(t) = \mathbb{E}\omega(t), \qquad \kappa(s, t)_{ij} = \mathrm{cov}(\omega(s)_i, \omega(t)_j).$$

The process is *stationary* if $\kappa(s, t) = k(t - s)$ for some $k : \mathbb{R} \to \mathbb{R}^{d \times d}$. For notational convenience, we focus on the zero-mean case, where $m \equiv 0$, though our results apply

to the general case as well. Zero-mean, stationary Gaussian processes are characterized entirely by the covariance function $k$.

We summarize the important properties of GAPs in the following theorem.

**Theorem 2.3** (Saatçi, 2012, Section 2.2). *Let $\omega$ be a zero-mean continuous, stationary Gaussian process on $\mathbb{R}^d$ with covariance function $k$. Then $\omega$ is a GAP of order $m$ if and only if its spectral density $S_\omega(\tau) := \frac{1}{2\pi} \int_{-\infty}^{\infty} k(t)e^{-it\tau}dt \in \mathbb{C}^{d\times d}$ is of the form*

$$S_\omega(\tau) = \sigma\sigma^\tau/f(\tau^2), \qquad \forall \tau \in \mathbb{R}$$

*where $f$ is a polynomial of degree $m$.*

*Moreover, if $\omega$ is a GAP of order $m$, then $\eta := (\omega, d\omega/dt, \ldots, d^{m-1}\omega/dt^{m-1})$ is a zero-mean, stationary Gauss–Markov process.*

With Theorem 2.3 in hand, we easily obtain many examples of GAPs. Crucially, this includes Gaussian processes corresponding to the Matérn kernel, the most popular smooth Gaussian process in applications, defined by the covariance function

$$k(t) \propto \sigma^2(\sqrt{2\nu}t/\ell)^\nu K_\nu(\sqrt{2\nu}t/\ell), \qquad (5)$$

where $K_\nu$ is the modified Bessel function of the second kind and $\nu$ and $\ell$ are positive parameters.

**Proposition 2.4** (Hartikainen & Särkkä, 2010, Section 4.1). *Suppose $\omega$ is a Gaussian process on $C(\mathbb{R}; \mathbb{R})$ whose covariance function is a Matérn kernel for smoothness parameter $\nu = m - 1/2$, for $m \in \mathbb{N}$. Then $\omega$ is a GAP of order $m$.*

By Theorem 2.3, the same holds true in the multidimensional case if each coordinate is taken to be independent of Matérn covariance.

Though the above characterization is formulated for stationary processes, our algorithm also applies to the nonstationary case, for instance, to the case of integrated Brownian motion: $\frac{d^m}{dt^m}\omega(t) = \sigma\xi(t)$. Taking $m = 1$ recovers the standard Schrödinger bridge, and $m = 2$ gives rise to the so-called "momentum Schrödinger bridge," previously studied in (Chen, Conforti, Georgiou, and Ripani, 2019; Chen, Liu, Tao, and Theodorou, 2023). The integrated Brownian motion prior possesses close connections to spline regression; see (Saatçi, 2012, Section 2.2.3) for more details.

## 3. Lifting Schrödinger Bridges

The remainder of this paper is devoted to giving an efficient algorithm for smooth SBs whose reference process is a GAP. In this section, we leverage the structure of GAPs to *lift* (3) to a higher-dimensional problem, with better structure. In

Section 4, we show that this reformulated problem can be solved by a belief propagation algorithm in a number of iterations that scales *linearly* with $K$. Finally, in Section 5, we develop a practical approximation of the belief propagation algorithm, with overall runtime quadratic in $K$.

In what follows, for notational convenience, we focus on the case $d = 1$. (We discuss the runtime considerations assciated with larger $d$ in Section 6.)

Let $R$ be the law of a mean zero GAP $\omega$ of order $m$ for an integer $m > 0$. Theorem 2.3 guarantees that $\eta := (\omega, d\omega/dt, \ldots, d^{m-1}\omega/dt^{m-1})$ is a stationary Gauss–Markov process on $\mathbb{R}^m$, whose law we denote by $\tilde{R}$. We write $\tilde{R}_{[K]}$ for the joint law of the finite-dimensional vector $\boldsymbol{\eta} := (\eta_0, \ldots, \eta_K)$, which is Gaussian with mean $0$ and covariance matrix $\tilde{\Sigma} \in \mathbb{R}^{m(K+1)\times m(K+1)}$.

We first show that the smooth Schrödinger bridge problem corresponding to $R$ can be rewritten in terms of $\tilde{R}$. Suppose for concreteness that $\mu_k$ is supported on a finite set $\mathcal{X}_k \subseteq \mathbb{R}$, and write $\mathcal{X}_{[K]} = \prod_{k=0}^{K} \mathcal{X}_k$. We write $\mathcal{Y}_{[K]} := \mathbb{R}^{(m-1)(K+1)}$. Given a probability measure $\tilde{P}$ on phase space $\mathcal{Z}_{[K]} := \mathcal{X}_{[K]} \times \mathcal{Y}_{[K]}$ whose marginal distribution on $\mathcal{Y}_{[K]}$ is absolutely continuous, we write $p(\mathbf{x}, \mathbf{y})$ for its density with respect to the product of the counting measure on $\mathcal{X}_{[K]}$ and the Lebesgue measure on $\mathcal{Y}_{[K]}$. We will use the variable $\mathbf{z} = (\mathbf{x}, \mathbf{y})$ to denote an element of $\mathcal{Z}_{[K]}$.

We obtain the following:

**Lemma 3.1.** *Assume that each of the marginals $\mu_k$ is supported on a finite set $\mathcal{X}_k$. There is a one-to-one correspondence between solutions to (3) and solutions to*

$$\min_p \frac{1}{2} \int_{\mathcal{Y}_{[K]}} \sum_{\mathbf{x}\in\mathcal{X}_{[K]}} \mathbf{z}^\top \tilde{\Sigma}^{-1}\mathbf{z}\, p(\mathbf{z})\, d\mathbf{y} - \mathcal{H}(p),$$

$$p_{x_k} = \mu_k, \quad \forall k \in [K], \qquad (6)$$

*where $\mathcal{H}(p) := -\int_{\mathcal{Y}_{[K]}} \sum_{\mathbf{x}\in\mathcal{X}_{[K]}} p(\mathbf{z})\log p(\mathbf{z})\, d\mathbf{y}$ and the minimization is taken over all densities on $\mathcal{Z}_{[K]}$ under which the marginal law of $x_k$ is equal to $\mu_k$.*

We refer to (6) as the *lifted* Schrödinger Bridge problem, since it is obtained by lifting the optimization problem from densities on $\mathcal{X}_{[K]}$ to densities on $\mathcal{Z}_{[K]}$.

At first sight, the lifted problem (6) is no improvement over (3)—indeed, the situation seems to have become worse due to the introduction of the continuous variables $\mathbf{y}$. However, we now show that unlike (3), problem (6) is directly amenable to efficient algorithms.

The first step is to leverage the Gauss–Markov property of $\tilde{R}_{[K]}$ to simplify (6). Recall the fundamental fact that, under $\tilde{R}$, the law of $\boldsymbol{\eta}$ has the Markov property. Given $\mathbf{z} = (\mathbf{x}, \mathbf{y}) \in \mathcal{Z}_{[K]}$ and $k \in [K]$, we write $\mathbf{z}_k = (x_k, \mathbf{y}_k)$

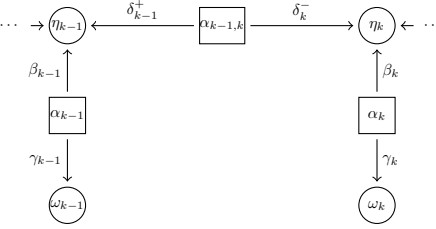

*Figure 3.* A portion of the graphical model corresponding to the joint law of $\boldsymbol{\omega}$ and $\boldsymbol{\eta}$.

for the $k$th coordinate of $\mathbf{z}$, which takes values in $\mathcal{Z}_k$. The Markov property then implies that $\mathbf{z}_{k+1}$ is independent of $\mathbf{z}_0, \ldots, \mathbf{z}_{k-1}$, conditioned on $\mathbf{z}_k$. At the level of densities, this implies the decomposition

$$
\begin{aligned}
r(\mathbf{z}) &= r(\mathbf{z}_0) \prod_{k=1}^{K} r(\mathbf{z}_k | \mathbf{z}_{k-1}) \\
&\propto r(\mathbf{z}_0) \prod_{k=1}^{K} \exp\left(-\frac{1}{2} Q_k(\mathbf{z}_{k-1}, \mathbf{z}_k)\right),
\end{aligned} \tag{7}
$$

where $r(\mathbf{z}_0) \propto \exp(-\frac{1}{2}\mathbf{z}_0^\top \tilde{\Sigma}_{0,0}^{-1}\mathbf{z}_0)$ and each $Q_k$ is a quadratic function:

$$
Q_k(\mathbf{z}_{k-1}, \mathbf{z}_k) := (\mathbf{z}_k - A_k\mathbf{z}_{k-1})^T \Lambda_k^{-1}(\mathbf{z}_k - A_k\mathbf{z}_{k-1}),
$$

for matrices $A_k := \tilde{\Sigma}_{k,k-1}(\tilde{\Sigma}_{k-1,k-1})^{-1} \in \mathbb{R}^{m \times m}$ and $\Lambda_k := \tilde{\Sigma}_{k,k} - \tilde{\Sigma}_{k,k-1}(\tilde{\Sigma}_{k-1,k-1})^{-1}\tilde{\Sigma}_{k-1,k} \in \mathbb{R}^{m \times m}$. These representations follow directly from standard formulas for Gaussian conditioning, and admit explicit expressions in terms of the kernel $k$ of the GAP (Saatçi, 2012, Section 2.3.1).

This decomposition represents the first term in (6) as a "tree-structured cost" (Haasler et al., 2021a), therefore rendering the marginalization in (4) amenable to belief propagation methods. We develop such a method in the next section.

## 4. Belief Propagation

The goal of this section is to show that a *belief propagation* algorithm can be used to efficiently solve the lifted problem (6). Belief propagation (Kschischang et al., 2001; Pearl, 1982) is a canonical approach for performing inference in high-dimensional models whose densities possess simple factorizations, such as the one given in (7). The use of belief propagation algorithms for Sinkhorn-type problems is not new (Haasler et al., 2021a; Singh et al., 2022; Teh & Welling, 2001); however, we stress that their application to Schrödinger bridges with smooth priors is novel.

To develop our algorithm, we first reformulate (4) in the language of graphical models. (See Koller & Friedman, 2009; Wainwright & Jordan, 2007, for background.) The

probabilistic structure of $\boldsymbol{\omega}$ and the lifted process $\boldsymbol{\eta}$ under $\tilde{R}$ means that we can represent their joint distribution by a simple hidden Markov model (see Figure 3), with observed variables $\{\omega_k\}_{k=0}^K$ and hidden variables $\{\eta_k\}_{k=0}^K$. These correspond to *variable nodes* in Section 3, denoted by circles.

The dependencies among these variables nodes are represented by *factor nodes* (squares in Figure 3): the factor nodes $\alpha_{k-1,k}$ connecting $\eta_{k-1}$ and $\eta_k$ enforce the joint law of the pair $(\eta_{k-1}, \eta_k)$, and the factor nodes $\alpha_k$ connecting $\eta_k$ and $\omega_k$ enforce the deterministic requirement that the first coordinate of $\eta_k$ equals $\omega_k$. We associate to $\alpha_{k-1,k}$ the *factor node potential* $\Phi_k : \mathcal{Z}_{k-1} \times \mathcal{Z}_k$ given by

$$
\Phi_1(\mathbf{z}_0, \mathbf{z}_1) := \exp\left(-\frac{\mathbf{z}_0^\top \tilde{\Sigma}_{0,0}^{-1}\mathbf{z}_0 + Q_1(\mathbf{z}_0, \mathbf{z}_1)}{2}\right)
$$

$$
\Phi_k(\mathbf{z}_{k-1}, \mathbf{z}_k) := \exp\left(-\frac{Q_k(\mathbf{z}_{k-1}, \mathbf{z}_k)}{2}\right) \quad k > 1,
$$

so that

$$
r(\mathbf{z}) \propto \prod_{k=1}^{K} \Phi_k(\mathbf{z}_{k-1}, \mathbf{z}_k).
$$

To find the optimal solution $p^*$ to (6), we use a belief propagation algorithm (Algorithm 2). The algorithm iteratively updates "messages" traveling between factor and variable nodes, which are depicted in Figure 3. The "vertical" messages $\beta_k, \gamma_k$ are real-valued functions on $\mathcal{X}_k$ which encode information about the $\boldsymbol{\omega}$ marginals of $p^*$, while the "horizontal" messages $\delta_k^+, \delta_k^-$ are real-valued functions on $\mathcal{Z}_k$ which encode information about the joint distribution of $\eta_k$ and its neighbors $\eta_{k-1}$ and $\eta_{k+1}$ These messages are iteratively updated back and forth across the graph via the application of the following operators:

$$
\mathcal{I}_k(\delta_k^-, \delta_k^+)(x_k) := \int_{\mathcal{Y}_k} \delta_k^-(x_k, \mathbf{y}_k)\delta_k^+(x_k, \mathbf{y}_k)d\mathbf{y}_k
$$

$$
\mathcal{L}_{k-1}(\delta_k^+, \beta_k)(\mathbf{z}_{k-1}) :=
$$

$$
\sum_{x_k \in \mathcal{X}_k} \int_{\mathcal{Y}_k} \Phi_k(\mathbf{z}_{k-1}, \mathbf{z}_k)\delta_k^+(\mathbf{z}_k)\beta_k(x_k)d\mathbf{y}_k
$$

$$
\mathcal{R}_k(\delta_{k-1}^-, \beta_{k-1})(\mathbf{z}_k) :=
$$

$$
\sum_{x_{k-1} \in \mathcal{X}_{k-1}} \int_{\mathcal{Y}_{k-1}} \Phi_k(\mathbf{z}_{k-1}, \mathbf{z}_k)\delta_k^-(\mathbf{z}_{k-1})\beta_k(x_{k-1})d\mathbf{y}_{k-1}
$$

$$
\tag{8}
$$

Since these operators involve manipulating functions on the space $\mathcal{Z}_k$, they are not directly implementable. In this section, we regard these operators as single basic operators for the purpose of complexity analysis. We develop efficient techniques to bypass their direct evaluation in the next section.

The main result of this section is that Algorithm 2 implicitly implements the Sinkhorn algorithm (Algorithm 1) for the multi-marginal entropic optimal transport problem (3), with cost given by

$$\exp(-C(\mathbf{x})) = \int \cdots \int \prod_{k=1}^{K} \Phi_k(\mathbf{z}_{k-1}, \mathbf{z}_k) d\mathbf{y}_0 \cdots d\mathbf{y}_K .$$

(9)

This connection allows us to develop a rigorous convergence guarantee for Algorithm 2. Indeed, (9) implies that $\exp(-C(\mathbf{x}))$ is, up to a normalizing constant, the joint density of $R_{[K]}$, so that Algorithm 1, and hence Algorithm 2, solves the smooth Schrödinger Bridge problem.

---

**Algorithm 2** Belief Propagation with Continuous Massages

1: **Input:** Factor node potentials $\{\Phi_k\}_{k=0}^{K-1}$, distributions $\{\mu_k\}_{k=0}^{K}$ on $\{\mathcal{X}_k\}_{k=0}^{K}$.
2: Initialize $\ell = 0, \delta_K^+ \equiv 1, \delta_0^- \equiv 1$, and $\beta_k^{(0)} : \mathcal{X}_k \to \mathbb{R}^+$, $k \in [K]$ arbitrary
3: **while** not converged **do**
4:    $\ell \leftarrow \ell + 1$
5:    **for** $k = K, ..., 1$ **do** {/* left pass */}
6:       $\delta_{k-1}^+ \leftarrow \mathcal{L}_{k-1}(\delta_k^+, \beta_k^{(\ell-1)})$
7:    **end for**

8:    $\gamma_0^{(\ell)} \leftarrow \mathcal{I}_0(\delta_0^-, \delta_0^+)$
9:    $\beta_0^{(\ell)} \leftarrow \mu_0 \oslash \gamma_0^{(\ell)}$

10:    **for** $k = 1, ..., K$ **do** {/* right pass */}
11:       $\delta_k^- \leftarrow \mathcal{R}_k(\delta_{k-1}^-, \beta_{k-1}^{(\ell)})$
12:       $\gamma_k^{(\ell)} \leftarrow \mathcal{I}_k(\delta_k^-, \delta_k^+)$
13:       $\beta_k^{(\ell)} \leftarrow \mu_k \oslash \gamma_k^{(\ell)}$
14:    **end for**

15: **end while**
16: Return $(\beta_k^{(\ell)})_{k \in [K]}$

---

**Theorem 4.1.** *Algorithm 2 is equivalent to Algorithm 1 with cost given by* (9)*, in the sense that, if the initialization of Algorithm 2 and Algorithm 1 satisfy*

$$\beta_k^{(0)} = v_k^{(0)} \quad \forall k \in [K]$$

(10)

*then, for all $\ell \geq 0$ and all $k = 0, ..., K$, we have*

$$\gamma_k^{(\ell)} = \mathbf{S}_k^{(\ell)}, \beta_k^{(\ell)} = v_k^{(\ell)}.$$

(11)

*In particular, we have the following two consequences:*

*(1) Algorithm 2 achieves an $\varepsilon$-approximate solution for* (3) *in $\tilde{O}(KC_{\max}/\varepsilon^{-1})$ iterations, where $C_{\max} = \max_{\mathbf{x} \in \mathcal{X}_{[T]}} C(\mathbf{x})$ and $\tilde{O}(\cdot)$ hides polylogarithmic factors in the problem parameters.*

*(2) The solution of* (6) *is given by*

$$p(\mathbf{z}) \propto \prod_{k=1}^{K} \Phi_k(\mathbf{z}_{k-1}, \mathbf{z}_k) \prod_{k=0}^{K} \beta_k^*(x_k)$$

(12)

*where $(\beta_k^*)_{k \in [K]}$ is the fixed point of Algorithm* (2)*.*

The representation in (12) implies that the output of Algorithm 2 can be used to efficiently manipulate and sample from the solution to the smooth SB problem; see Appendix D for details.

## 5. Approximate Belief Propagation

The final ingredient of our algorithm is an approximation scheme to efficiently implement the operators in (8). We use the technique proposed by Noorshams & Wainwright (2013): we decompose the continuous messages in a suitable orthonormal basis. The orthonormal decomposition method provides two key benefits. First, by truncating the series at a sufficiently high order, we can create an accurate representation of the messages using only a finite number of orthogonal coefficients. Second, the key update rules in Algorithm 2 described in (8) involve taking the $L^2$ inner product of two continuous messages. Expressing the messages in an orthonormal basis simplifies these operations considerably.

Let $\{\phi_k^i\}_{i=0}^{\infty}$ represent an orthonormal basis in the space $L^2(\mathbb{R}^{m-1})$. The subscript $k$ indicates that one is free to choose different bases for different $k \in [K]$.

We express the horizontal messages $\delta_k^+$ and $\delta_k^-$ in terms of this basis as follows:

$$\delta_k^+(\mathbf{z}_k) = \sum_{i=1}^{\infty} \ell_k^i(x_k)\phi_k^i(\mathbf{y}_k),$$

(13)

$$\delta_k^-(\mathbf{z}_k) = \sum_{i=1}^{\infty} r_k^i(x_k)\phi_k^i(\mathbf{y}_k),$$

(14)

where $r$ and $\ell$ denote the coefficients for rightward and leftward messages and the coefficients are given by

$$\ell_k^i(x_k) := \int \delta_k^+(x_k, \mathbf{y}_k)\phi_k^i(\mathbf{y}_k)d\mathbf{y}_k$$

$$r_k^i(x_k) := \int \delta_k^-(x_k, \mathbf{y}_k)\phi_k^i(\mathbf{y}_k)d\mathbf{y}_k,$$

Utilizing the orthonormal expansions from (13)-(14), we can re-write Algorithm 2 so that it operates directly on the coefficients $\boldsymbol{l}_k := (\ell_k^i)_{i=0}^{\infty}$ and $\boldsymbol{r}_k := (r_k^i)_{i=0}^{\infty}$, which are functions from $\mathcal{X}_k$ to $\ell_2$.

First, by $L^2(\mathcal{Y}_k)$ orthogonality, the update rule for $\gamma_k$ can be written

$$\mathcal{I}_k(\delta_k^+, \delta_k^-)(x_k) = \langle \boldsymbol{l}(x_k), \boldsymbol{r}(x_k) \rangle.$$

Note that in this notation, we can write

$$\beta_k(x_k) = \mu_k(x_k)/\gamma_k(x_k) = \mu_k(x_k)\langle l(x_k), r(x_k)\rangle^{-1}$$

We now consider the update rule for the left and right messages. The following lemma shows how to express these updates in terms of coefficients.

**Lemma 5.1.** *Fix functions $\delta_k^+ = \sum_{i=1}^{\infty} \ell_k^i \otimes \phi_k^i$ and $\beta_k$. Then the coefficients of $\delta_{k-1}^+ = \mathcal{L}_{k-1}(\delta_k^+, \beta_k)$ in the basis $(\phi_{k-1}^i)_{i=1}^{\infty}$ are given by*

$$\ell_{k-1}^i(x_{k-1}) = \sum_{x_k \in \mathcal{X}_k} \beta_k(x_k) \sum_{j=1}^{\infty} \ell_k^j(x_k)\Gamma_{k-1}^{ij}(x_{k-1}, x_k), \tag{15}$$

*where $\Gamma_{k-1}^{ij}(x_{k-1}, x_k)$ denotes*

$$\iint \phi_{k-1}^i(\mathbf{y}_{k-1})\phi_k^j(\mathbf{y}_k)\Phi_k(\mathbf{z}_{k-1}, \mathbf{z}_k)d\mathbf{y}_{k-1}d\mathbf{y}_k. \tag{16}$$

*Similarly, if $\delta_{k-1}^- = \sum_{i=1}^{\infty} r_{k-1}^i \otimes \phi_{k-1}^i$, then the coefficients of $\delta_k^- = \mathcal{R}_k(\delta_{k-1}^-, \beta_{k-1})$ in the basis $(\phi_k^i)_{i=1}^{\infty}$ are given by*

$$r_k^i(x_k) =$$
$$\sum_{x_{k-1} \in \mathcal{X}_{k-1}} \beta_{k-1}(x_{k-1}) \sum_{j=1}^{\infty} r_{k-1}^j(x_{k-1})\Gamma_{k-1}^{ij}(x_{k-1}, x_k). \tag{17}$$

To obtain a practical procedure, we replace the infinite sums in (13)-(14) with finite approximations. Choosing a sufficiently large $M$, and using the first $M$ orthonormal functions on the basis to approximate $\delta_k^+$ and $\delta_k^-$, we repeat the previous steps to derive the update rules, expressed as matrix-vector multiplications.

$$\beta_k(x_k) \leftarrow \mu(x_k)(l_k(x_k)^T r_k(x_k))^{-1}, \tag{18}$$

$$l_{k-1}(x_{k-1}) \leftarrow \sum_{x_k \in \mathcal{X}_k} \beta_k(x_k)\Gamma_{k-1}(x_{k-1}, x_k)l_k(x_k) \tag{19}$$

$$r_k(x_k) \leftarrow \tag{20}$$
$$\sum_{x_{k-1} \in \mathcal{X}_{k-1}} \beta_{k-1}(x_{k-1})r_{k-1}^T(x_{k-1})\Gamma_{k-1}(x_{k-1}, x_k)$$

where $l_{k-1}(x_{k-1})$ and $r_k(x_k)$ are vectors in $\mathbb{R}^M$ and $\Gamma_{k-1}(x_{k-1}, x_k)$ is a matrix in $\mathbb{R}^{M \times M}$ whose element in row $i$ and column $j$ is given by $\Gamma_{k-1}^{ij}(x_{k-1}, x_k)$.

Upon convergence of this algorithm, we obtain the coefficients $r_k$ and $l_k$, and thereby obtain estimates of $\beta_k^*$. As discussed after Theorem 4.1, these messages can be used directly for downstream tasks involving the Schrödinger bridge.

---

**Algorithm 3** Approximate Belief Propagation Algorithm

Precompute $\mathbf{\Gamma}_0, ..., \mathbf{\Gamma}_{K-1} \in \mathbb{R}^{n \times n \times M \times M}$ by (16).
Initialize $\{r_k(x_k)\}_{k,x_k}$ and $\{l_k(x_k)\}_{k,x_k}$ as $M$ dimensional vectors filled with 1's
**while** not converged **do**
  **for** $k = K, ..., 1$ **do**
    Update $l_{k-1}$ by (19)
  **end for**
  Calculate $\beta_0$ by (18)
  **for** $k = 1, ..., K$ **do**
    Update $r_k$ by (20)
    Calculate $\beta_k$ by (18)
  **end for**
**end while**
Return $r_k$ and $l_k$

---

# 6. Time complexity and practical considerations

Implementing Algorithm 3 requires selecting bases $(\{\phi_k^i\}_{i=1}^{\infty})_{k \in [K]}$ along with a number of coefficients $M$. The computational complexity of the resulting algorithm scales directly with $M$, which we summarize in the following result.

**Theorem 6.1.** *Executing $T$ iterations of Algorithm 3 takes $O(TKn^2M^2)$ time. In particular, executing $T = \tilde{O}(KC_{\max}\varepsilon^{-1})$ iterations takes $\tilde{O}(K^2n^2M^2C_{\max}\varepsilon^{-1})$ time.*

Theorem 4.1 suggests that $T = \tilde{O}(KC_{\max}\varepsilon^{-1})$ iterations suffice to obtain a $\varepsilon$-approximate solution to (6); unfortunately, however, we lack a rigorous approximation guarantee quantifying the difference between the output of Algorithm 3 and that of Algorithm 2. Nevertheless, the success of our empirical results (Section 7) indicates that Algorithm 3 does offer a good approximation for the solution to the smooth SB problem. We leave the open question of demonstrating this fact theoretically to future work.

The main tuning parameter of our algorithm is the choice of $M$. In principle, this choice should depend on the smoothness of the messages $\delta_k^+$ and $\delta_k^-$, the order $m$ of the GAP, and the dimension $d$. Since $\delta_k^+$ and $\delta_k^-$ correspond to Gaussian densities, their expansion in many reasonable bases (for example, Fourier or Wavelet bases) will exhibit strong decay; however, since they are defined on $\mathcal{Y}_k = \mathbb{R}^{(m-1)d}$, standard smoothness arguments would predict that the necessary number of coefficients scales exponentially with the product $md$.

However, the dependence on $d$ can be somewhat ameliorated under additional assumptions. Suppose that the GAP has independent coordinates and the orthogonal bases have tensor product structure, so that each basis element

$\phi \in L^2(\mathbb{R}^{(m-1)d})$ satisfies $\phi(\mathbf{y}) = \prod_{j=1}^d \varphi^{(j)}(\mathbf{y}^{(j)})$ for $\varphi^{(j)} \in L^2(\mathbb{R}^{m-1})$ and where $\mathbf{y}^{(j)}$ corresponds to the derivatives of the $j$th coordinate of $\omega$. In this case, the tensor $\mathbf{\Gamma}$ factors across each dimension into the tensor product of $d$ smaller tensors $\tilde{\Gamma}^{(j)}$,$(j = 1, ..., d)$. Suppose the number of coefficients we use is equal for each dimension, i.e. $\tilde{\Gamma}^{(j)} \in \mathbb{R}^{n \times n \times M^{\frac{1}{d}} \times M^{\frac{1}{d}}}$, then the complexity of steps (19) and (20) drops to $O(dn^2 M^{1+\frac{1}{d}})$ by taking advantage of this lower rank structure. In this important special case, therefore, the time complexity of our algorithm scales more benignly with $d$, which allows us to take larger $M$ when $d$ rises. A record of the running time of one iteration of message passing against $M$ and dimension $d$ is presented in Table 9.

# 7. Experiments

We test our algorithm on two kinds of low-dimensional smooth trajectory inference tasks. The first kind aims at tracking the exact trajectory of each individual particle (e.g. Figure 2), which we refer to as the One-By-One (OBO) tracking task in the following text. For this kind of task, we evaluate the performance by calculating the distance of each inferred trajectory and the ground truth and measure the percentage of time that the algorithm tracks a particle correctly. For the second kind, the task is to infer the group trajectories of point clouds whose evolution has geometric structure. We provide two ways to evaluate the performance for this kind of task, similar to the evaluation in Banerjee et al.. We first keep all the observations and visualize the trajectories and see if they form a pattern that is close to the ground-truth pattern. Secondly, we will leave out observations at a certain timestep and instead infer the position of particles at this timestep and evaluate the distance between the inferred and real observations, which we call Leave-One-Timestep-Out (LOT) tasks. Our code for reproducing these experiments is available on Github.[1]

## 7.1. One-By-One tracking

For OBO tasks, we consider three synthetic datasets where trajectories of particles intersect frequently. We compare our algorithm with the standard Schrödinger Bridge (SB) and a modification based on computing the $W_2$ optimal matching at each step (W2M). We test on three data sets, two 2-dimensional data sets (Fig 7 in Appendix F) and a 3-dimensional data set consisting of the simulated orbits of an $N$-body physical system (Fig. 4). Smooth SB performs second-best on the Tri-stable diffusion dataset and substantially outperforms other approaches on the more challenging $N$-Body and Gaussian Process data. Quantitative evaluations are summarized in Table 1. Full experimental details

[1] https://github.com/WanliHongC/Smooth_SB

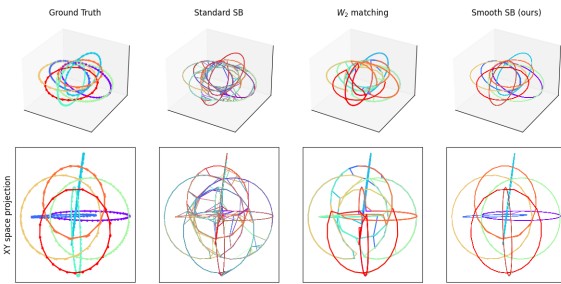

*Figure 4.* Visualization of orbits in 3D space. The colors of trajectories depend on the starting point. The second row is the visualization of the XY-space projection of the corresponding above plot.

*Table 1.* A comparison between smooth SB and two baseline particle tracking methods. Information on evaluation metrics appears in Appendix F.

| Dataset | Method | JumpP | 5p acc | Mean $\ell_2$ |
|---|---|---|---|---|
| Tri-stable Diffusion | SSB (ours) | 1.12e-1 | 0.798 | 1.42e-2 |
| | BMSB | 5.20e-1 | 0.171 | 6.79e-1 |
| | W2M | **2.25e-2** | **0.956** | **1.37e-8** |
| N Body | SSB (ours) | **5.00e-4** | **0.999** | **5.48e-4** |
| | BMSB | 1.14e-1 | 0.641 | 5.43e-1 |
| | W2M | 1.08e-1 | 0.649 | 5.50e-1 |
| 2D Gaussian Process | SSB (ours) | **5.60e-3** | **0.991** | **3.57e-4** |
| | BMSB | 1.37e-1 | 0.612 | 2.82e-1 |
| | W2M | 6.60e-2 | 0.760 | 2.32e-1 |

appear in Appendix F.

## 7.2. Point clouds trajectory inference

We also test our algorithm on five challenging baselines in the trajectory inference literature. We compare our algorithm quantitatively on tasks of LOT with three other state-of-the-art algorithms: MIOFlow (Huguet et al., 2022b), DMSB (Chen et al., 2023) and F&S (Chewi et al., 2021). A visualization of the output of our algorithm appears in Figure 5, 6 and the quantitative results for the LOT tasks are provided in Table 2. Our algorithm can recover the geometric pattern in each low-dimensional dataset by generating smooth trajectories. We can also apply our algorithm to somewhat higher dimensional settings by appropriately tuning the number of coefficients in the orthonormal expansion. In particular, in our experiments on the 10-dimensional Dyngen cycle dataset shown in Figure 6, we use 4 approximation coefficients for each of the first 5 dimensions and 1 approximation coefficient for the each of the last 5 dimensions. Despite the small number of coefficients, the resulting trajectories are still meaningful. Full experimental details appear in Appendix F.

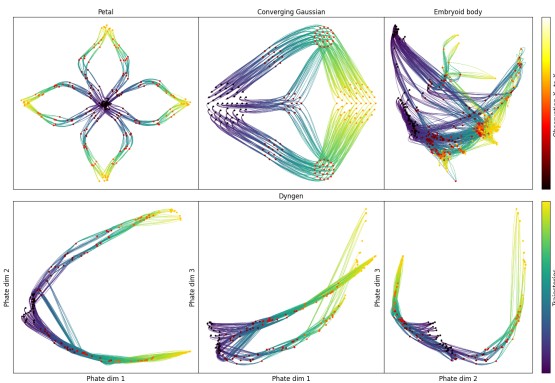

*Figure 5.* Visualization of trajectory inference on various datasets by lifted SB, for the 5D Dyngen Tree data, we visualize the 2D projection of the first three dimensions in the second row.

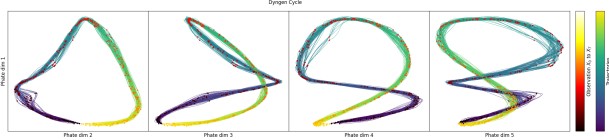

*Figure 6.* Visualization of trajectory inference on the 10D Dyngen cycle dataset projected by Phate, we visualize the 2D projection of the first dimension against the second to fifth dimension.

*Table 2.* Performance comparison on the Leave-One-Timestep-Out task at step $j$ between our algorithm and the state-of-art algorithms. Our algorithm is typically **best** or *second-best*. DMSB failed to converge on the Dyg dataset despite extensive tuning. Information on evaluation metrics appears in Appendix F.

| Dataset | Method | $\mathbf{W}_1(\downarrow)$ | $\mathbf{M}_G(\downarrow)$ | $\mathbf{M}_I(\downarrow)$ |
|---|---|---|---|---|
| Petal $j=2$ | SSB (ours) | *2.70e-2* | **2.21e-5** | **3.75e-3** |
| | MIOFlow | 2.22e-1 | 9.06e-3 | 1.11e-1 |
| | DMSB | 2.10e-1 | 5.52e-3 | 3.68e-2 |
| | F&S | **2.05e-2** | *2.85e-5* | *4.48e-3* |
| EB $j=2$ | SSB (ours) | **8.45e-2** | 2.46e-3 | 5.04e-2 |
| | MIOFlow | 1.34e-1 | **1.36e-3** | **2.81e-2** |
| | DMSB | 1.46e-1 | 9.43e-3 | 9.72e-2 |
| | F&S | *8.72e-2* | *1.47e-3* | *3.88e-2* |
| Dyg Tree $j=1$ | SSB (ours) | *9.81e-2* | **1.64e-3** | **3.51e-2** |
| | MIOFlow | 2.33e-1 | 2.82e-2 | 1.73e-1 |
| | DMSB | * | * | * |
| | F&S | **9.78e-2** | *2.00e-3* | *4.64e-2* |
| Dyg Cycle $j=7$ | SSB (ours) | **1.94e-1** | **2.85e-2** | *1.71e-1* |
| | MIOFlow | 4.25e-1 | 1.57e-1 | 4.11e-1 |
| | DMSB | * | * | * |
| | F&S | *2.84e-1* | *2.91e-2* | **1.70e-1** |

## 8. Discussions and Future Directions

We have presented a new method for trajectory inference and particle tracking based on smooth Schrödinger bridges, which achieves very good performance on a number of challenging benchmarks. The main limitations of our proposal are related to the approximate implementation developed in Section 5. As we have discussed, our proposal suffers from the curse of dimensionality, because the number of coefficients $M$ typically scales exponentially with respect to both the order of the GAP ($m$) and the dimension of the observations ($d$). In numerical experiments, relatively small values of $M$ (of order 1000) seem to perform well for problems up to dimension 10. An important question for future work is to either develop an approach with better dimensional scaling or, alternatively, show that the exponential scaling in dimension is unavoidable, as is the case for Wasserstein barycenters (Altschuler & Boix-Adserà, 2022).

Implementing our approach also requires selecting a suitable GAP to use as a reference process. As our experiments make clear, it is not necessary that the reference process match the data generating process precisely (see, e.g., Figure 2). However, picking an appropriate variance for the Gaussian process is important for good performance (see Figure 16). In our experiments, choosing $\sigma \approx \sigma_{\mathrm{data}}$ where $\sigma_{data}$ is a diagonal matrix containing the empirical standard deviation along each dimension, typically works well.

Finally, we have considered a definition of the Schrödinger bridge which enforces the strict marginal constraint $P_k = \mu_k$. In applications, it is natural to assume that observations of the particles are corrupted with noise, which motivates a version of the SB with an approximate constraint $P_k \approx \mu_k$ (Chizat et al., 2022; Lavenant et al., 2024). It is possible to incorporate noisy observations into the graphical model framework we describe above by introducing a suitable potential at the factor nodes $\alpha_k$. A similar approach generalizes to other missing data problems, for instance, when some dimensions are not observed. We leave this extension to future work.

## Acknowledgements

JNW acknowledges the support of National Science Foundation grant DMS-2339829.

## Impact Statement

This paper presents work whose goal is to advance the field of Machine Learning. There are many potential societal consequences of our work, none which we feel must be specifically highlighted here.

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

## A. List of notation

- $\mathcal{P}(\mathcal{X})$: the set of Borel probability measures on $\mathcal{X}$.

- $D(\cdot|\cdot)$: Kullback–Leibler divergence. For probability measures $P$ and $R$,

$$D(P|R) := \begin{cases} \int \log \frac{dP}{dR}\, dP & P \ll R \\ +\infty & \text{otherwise.} \end{cases}$$

- $C(\mathbb{R}; \mathbb{R}^d)$, $C^m(\mathbb{R}; \mathbb{R}^d)$: continuous (respectively, $m$-times continuously differentiable) functions from $\mathbb{R}$ to $\mathbb{R}^d$.

- $[N], [N]^+$: for a positive integer $N$, $[N]$ is the set $0, ..., N$ while $[N]^+$ is the set $1, ..., N$.

- $\mathbb{R}^+$: the set of positive real numbers

- $\mathcal{X}_k$: the support of $\mu_k$, assumed to be finite, $\mathcal{Y}_k$: the space $\mathbb{R}^{(m-1)d}$, identified with the possible values of $(d\omega/dt, \ldots, d^{m-1}\omega/dt^{m-1})$ at $t = t_k$, $\mathcal{Z}_k$: the phase space $\mathcal{X}_k \times \mathcal{Y}_k$.

- $\mathcal{X}_{[T]} = \prod_{k \in [T]} \mathcal{X}_k$, $\mathcal{Y}_{[T]} = \prod_{k \in [T]} \mathcal{Y}_k$, $\mathcal{Z}_{[T]} = \prod_{k \in [T]} \mathcal{Z}_k$.

- $\mathbf{1}_{o_k}(x_k)$: the indicator function $\mathbf{1}_{o_k}(x_k) := \begin{cases} 1 & \text{if } x_k = o_k \\ 0 & \text{otherwise.} \end{cases}$

- $\odot$ and $\oslash$: point-wise multiplication and division

- $\text{Poly}(K_1, \ldots, K_\ell)$: a (multivariate) polynomial function in parameters $K_1, \ldots, K_\ell$.

## B. Prior work

There has been significant prior work on both the Schrödinger bridge problem and its applications in trajectory inference. Here, we briefly survey some important related contributions.

The Schrödinger bridge has been the subject of significant theoretical interest since its introduction; see (Léonard, 2014) for historical details and additional context. Currently, there are many methodological approaches to the Schrödinger bridge problem (see, e.g. De Bortoli et al., 2021; Gushchin et al., 2023; Pavon et al., 2021; Pooladian & Niles-Weed, 2024), almost all of which focus on the two-marginal case (in our notation $K = 1$). As was forcefully pointed out by Altschuler & Boix-Adserà, the multi-marginal case presents significantly greater computational challenges (Altschuler & Boix-Adserà, 2022; 2023); indeed, no general-purpose algorithms with running time polynomial in $K$ exist. However, it is possible to obtain polynomial-time algorithms in certain special cases (Altschuler & Boix-Adserà, 2023). Our work shows that smooth Schrödinger bridges with GAP priors are such a case.

The connection between multi-marginal entropic OT problems and belief propagation has been highlighted in a number of prior works dating back more than two decades (Singh et al., 2022; Haasler et al., 2021b; Teh & Welling, 2001), though apparently without connection to smooth Schrödinger Bridge problems. The exception is the "momentum Schrödinger bridge" of (Chen, Conforti, Georgiou, and Ripani, 2019; Chen, Liu, Tao, and Theodorou, 2023), which is, implicitly, an example of a smooth Schrödinger with prior given by integrated Brownian motion; the Sinkhorn algorithm proposed in (Chen et al., 2019) can be viewed as an *ad hoc* version of belief propagation for this special case. However, those works do not make the connection with smooth Gaussian processes, and their algorithm does not apply to more general GAPs.

The importance of the Schrödinger bridge problem for trajectory inference was implicitly recognized in the pioneering work (Schiebinger et al., 2019), and developed mathematically by (Chizat et al., 2022; Lavenant et al., 2024). The fact that the vanilla Shrödinger bridge problem uses a Markov prior is crucial for algorithms (Chizat et al., 2022), but has also been recognized as an undesirable feature that has led to the development of different trajectory inference methods which give rise to smoother paths. Apart from (Shen et al.), which proposes a "robust" version of the Schrödinger Bridge problem in which a single reference process is replaced by a family of (Markov) reference processes, these alternate methods largely abandon Schrödinger's formulation and develop very different techniques. Many of the most successful methods are based on measure-valued generalizations of splines (Banerjee et al.; Benamou et al., 2019; Chen et al., 2018; Chewi et al., 2021; Clancy & Suarez, 2022; Justiniano et al., 2024). Other methods based on neural networks have also been proposed (Huguet et al., 2022a; Tong et al., 2020). We compare these methods in our experimental section.

## C. Additional results and omitted proofs

The following lemma shows that any Markov Gaussian process with differentiable sample paths is essentially trivial.

**Lemma C.1.** *Let $\omega : \mathbb{R}^+ \to \mathbb{R}$ be a real-valued Gaussian process that is Markovian and almost surely differentiable. Assume $\mathrm{Var}[\omega(t)] > 0$ for all $t \geq 0$. Then $\omega(t) = \mathbb{E}[\omega(t)|\omega(0)]$ for all $t \geq 0$ almost surely.*

*Proof of Lemma C.1.* Since $\omega$ is an almost surely differentiable Gaussian process, we know that the covariance kernel $K(t,s) := \mathbb{E}[\omega(t)\omega(s)] = \mathrm{Cov}(\omega(t), \omega(s))$ is differentiable in both coordinates and $\frac{\partial^2}{\partial x_1 \partial x_2} K(t,s)$ exists. It is well known (see (Borisov, 1983)) that a real-valued Gaussian process is Markovian if and only if, for any $t_1 \leq t_2 \leq t_3$, we have

$$K(t_1, t_2)K(t_2, t_3) = K(t_1, t_3)K(t_2, t_2). \tag{21}$$

Let $t \leq s$ and $\varepsilon > 0$. From (21), we have $K(s,s)K(t, s+\varepsilon) = K(t,s)K(s, s+\varepsilon)$. Dividing each side by $\varepsilon$ and taking the limit as $\varepsilon \to 0$, we obtain

$$K(s,s)\frac{\partial}{\partial x_2}K(t,s) = K(t,s)\frac{\partial}{\partial x_2}K(s,s). \tag{22}$$

Similarly, we also have $K(t,t)\frac{\partial}{\partial x_1}K(t,s) = K(t,s)\frac{\partial}{\partial x_1}K(t,t)$. Combining these two identities, it follows that

$$K(t,t)K(s,s)\frac{\partial^2}{\partial x_1 \partial x_2}K(t,s) = K(t,s)\frac{\partial}{\partial x_1}K(t,t)\frac{\partial}{\partial x_2}K(s,s). \tag{23}$$

Note that $\frac{\partial}{\partial x_1}K(t,s) = \mathrm{Cov}(\omega'(t), \omega(s))$ and $\frac{\partial^2}{\partial x_1 \partial x_2}K(t,s) = \mathrm{Cov}(\omega'(t), \omega'(s))$ by interchanging differentiation and expectation.

For any $s$, by taking $t = s$ in (23), we obtain

$$\mathrm{Var}(\omega'(s))\mathrm{Var}(\omega(s)) = \mathrm{Cov}(\omega'(s), \omega(s))^2, \tag{24}$$

which further implies that

$$\mathrm{Var}(\omega'(s)|\omega(s)) = \mathrm{Var}(\omega'(s)) - \frac{\mathrm{Cov}(\omega'(s), \omega(s))^2}{\mathrm{Var}(\omega(s))} = 0.$$

Let $g_s : [0, s] \mapsto \mathbb{R}^+$ be given by

$$g_s(t) := \mathrm{Var}(\omega'(s)|\omega(t)). \tag{25}$$

We have shown that $g_s(s) = 0$. Moreover, for any $t \in [0, s)$, we have

$$g_s'(t) = \frac{d}{dt}\left(\mathrm{Var}(\omega'(s)) - \frac{\mathrm{Cov}(\omega'(s), \omega(t))^2}{\mathrm{Var}(\omega(t))}\right) \tag{26}$$

$$= \frac{2}{K(t,t)^2}\left(\frac{\partial}{\partial x_1}K(t,t) \cdot \frac{\partial}{\partial x_2}K(t,s)^2 - \frac{\partial}{\partial x_2}K(t,s) \cdot \frac{\partial^2}{\partial x_1 \partial x_2}K(t,s) \cdot K(t,t)\right) \tag{27}$$

$$= \frac{2}{K(t,t)^2}\left(\frac{\partial}{\partial x_1}K(t,t) \cdot \frac{\partial}{\partial x_2}K(t,s)^2 - \frac{\partial}{\partial x_2}K(t,s) \cdot K(t,s) \cdot \frac{\partial}{\partial x_1}K(t,t) \cdot \frac{\frac{\partial}{\partial x_2}K(s,s)}{K(s,s)}\right) \tag{28}$$

$$= \frac{2}{K(t,t)^2}\left(\frac{\partial}{\partial x_1}K(t,t) \cdot \frac{\partial}{\partial x_2}K(t,s)^2 - \frac{\partial}{\partial x_2}K(t,s)^2 \cdot \frac{\partial}{\partial x_1}K(t,t)\right) = 0, \tag{29}$$

where the third line uses (23) and the last line uses (22). Consequently, we have shown that $\mathrm{Var}(\omega'(s)|\omega(t)) = 0$ for any $t < s$. We complete the proof by applying the Cauchy–Schwarz inequality: for any $s > 0$, we have

$$\mathrm{Var}\left(\omega(s)|\omega(0)\right) = \mathrm{Var}\left(\omega(s) - \omega(0) \,|\, \omega(0)\right) \leq s \int_0^s \mathrm{Var}\left(\omega'(\ell)|\omega(0)\right) d\ell = 0. \tag{30}$$

Therefore, $\omega(s) = \mathbb{E}[\omega(s)|\omega(0)]$ almost surely for all $s > 0$.

$\square$

## C.1. Proof of Lemma 2.1

This follows directly from known arguments in the two-marginal case (see Léonard, 2014), which we recapitulate below.

We first show the equivalence between (1) and (2), which requires no assumptions on the marginal measures. We can decompose the measure $R$ as $R(\omega) = R(\omega \mid \boldsymbol{\omega}) R_{[K]}(\boldsymbol{\omega})$, where $R(\omega \mid \boldsymbol{\omega})$ denotes the conditional law of $\omega$ given the values $\boldsymbol{\omega} = (\omega_0, \ldots, \omega_K)$, and similarly for $P$. By the chain rule for Kullback–Leibler divergence, we obtain

$$\mathrm{D}(P \mid R) = \mathrm{D}(P_{[K]} \mid R_{[K]}) + \mathbb{E}_{\boldsymbol{\omega} \sim P_{[K]}} \mathrm{D}(P(\cdot \mid \boldsymbol{\omega}) \mid R(\cdot \mid \boldsymbol{\omega})).$$

For any choice of $P_{[K]}$, choosing $P(\omega \mid \boldsymbol{\omega}) = R(\omega \mid \boldsymbol{\omega})$ makes the second term vanish. Therefore, any solution $P^*$ of (1) must be of the form $P^*_{[K]}(\omega) R(\omega \mid \boldsymbol{\omega})$ for a solution $P^*_{[K]}$ of (2), and conversely.

For the second claim, we assume that each of the measures $\mu_k$ is absolutely continuous with finite entropy, and that $R_{[K]}$ has a density. In this case, any feasible solution to (2), i.e., any $P_{[K]}$ such that $\mathrm{D}(P_{[K]} \| R_{[K]}) < \infty$, must be absolutely continuous with respect to $R_{[K]}$ and hence have a density as well, which we also denote by $P_{[K]}$. We obtain

$$D(P_{[K]} \mid R_{[K]}) = \int \log \frac{P_{[K]}(\boldsymbol{\omega})}{\exp(-C(\boldsymbol{\omega}))} P_{[K]}(\boldsymbol{\omega}) \, d\boldsymbol{\omega}$$

$$= \int C(\omega) P_{[K]}(\boldsymbol{\omega}) \, d\boldsymbol{\omega} + \int P_{[K]}(\boldsymbol{\omega}) \log \frac{P_{[K]}(\boldsymbol{\omega})}{\prod_{k \in [K]} \mu_k(\omega_k)} \, d\boldsymbol{\omega} + \int P_{[K]}(\boldsymbol{\omega}) \log \prod_{k \in [K]} \mu_k(\omega_k) \, d\boldsymbol{\omega}.$$

The marginal constraints imply that

$$\int P_{[K]}(\boldsymbol{\omega}) \log \prod_{k \in [K]} \mu_k(\omega_k) \, d\boldsymbol{\omega} = \sum_{k \in [K]} \int \mu_k(\omega_k) \log \mu_k(\omega_k) \, d\omega_k.$$

Since each $\mu_k$ has finite entropy by assumption, this sum is finite and constant over the constraint set, so it can be dropped from the objective without changing the solutions. $\qquad\square$

## C.2. Proof of Lemma 3.1

Lemma 3.1 can be reformulated as follows: there is a one-to-one correspondence between solutions to

$$\max_{P_{[K]}} \sum_{\mathbf{x} \in \mathcal{X}_{[K]}} P_{[K]}(\mathbf{x}) \log(r(\mathbf{x})) - \sum_{\mathbf{x} \in \mathcal{X}_{[K]}} P_{[K]}(\mathbf{x}) \log \left( P_{[K]}(\mathbf{x}) / \prod_{k \in [K]} \mu_k(x_k) \right) \tag{31}$$
$$P_k = \mu_k, \forall k \in [K]$$

and

$$\max_p \int_{\mathcal{Y}_{[K]}} \sum_{\mathbf{x} \in \mathcal{X}_{[K]}} p(\mathbf{z}) \log(r(\mathbf{z})) \, d\mathbf{y} - \int_{\mathcal{Y}_{[T]}} \sum_{\mathbf{x} \in \mathcal{X}_{[T]}} p(\mathbf{z}) \log(p(\mathbf{z})) d\mathbf{y} \tag{32}$$
$$p_{x_k} = \mu_k \, \forall k \in [T],$$

where the maximization in (31) is taken over distributions in $\mathcal{P}(\mathcal{X}_{[T]})$ and the maximization in (32) is taken over probability densities on $\mathcal{Z}_{[K]}$.

First, note for any $P_{[T]}$ satisfying the marginal constraints $P_k = \mu_k$ for all $k \in [K]$,

$$\sum_{\mathbf{x} \in \mathcal{X}_{[K]}} P_{[K]}(\mathbf{x}) \log \prod_{k \in [K]} \mu_k(x_k) = \sum_{k \in [K]} \sum_{\mathbf{x} \in \mathcal{X}_{[K]}} P_{[K]}(\mathbf{x}) \log \mu_k(x_k) = \sum_{k \in [K]} \sum_{x_k \in \mathcal{X}_k} \mu_k(x_k) \log \mu_k(x_k).$$

Therefore, the term $\sum_{\mathbf{x} \in \mathcal{X}_{[K]}} P_{[K]} \log \prod_{k \in [K]} \mu_k(x_k)$ in (31) is constant on the feasible set and can be dropped from the objective.

Next, by the law of total probability,

$$\int_{\mathcal{Y}_{[T]}} \sum_{\mathbf{x} \in \mathcal{X}_{[T]}} p(\mathbf{z}) \log(r(\mathbf{z})) \, d\mathbf{y} = \int_{\mathcal{Y}_{[T]}} \sum_{\mathbf{x} \in \mathcal{X}_{[T]}} p(\mathbf{z}) \log(r(\mathbf{x})) \, d\mathbf{y} + \int_{\mathcal{Y}_{[T]}} \sum_{\mathbf{x} \in \mathcal{X}_{[T]}} p(\mathbf{z}) \log(r(\mathbf{y}|\mathbf{x})) \, d\mathbf{y} \tag{33}$$

and

$$\int_{\mathcal{Y}_{[T]}} \sum_{\mathbf{x} \in \mathcal{X}_{[T]}} p(\mathbf{z}) \log(p(\mathbf{z})) d\mathbf{y} = \int_{\mathcal{Y}_{[T]}} \sum_{\mathbf{x} \in \mathcal{X}_{[T]}} p(\mathbf{z}) \log(p(\mathbf{x})) d\mathbf{y} + \int_{\mathcal{Y}_{[T]}} \sum_{\mathbf{x} \in \mathcal{X}_{[T]}} p(\mathbf{z}) \log(p(\mathbf{y}|\mathbf{x})) \, d\mathbf{y}. \tag{34}$$

where $r(\mathbf{x})$ and $r(\mathbf{y}|\mathbf{x})$ denotes the marginal density of the $\mathbf{x}$ variables and conditional density of the $\mathbf{y}$ variables under $\tilde{R}_{[T]}$, and analogously for $p$. It follows from exchanging the order of integration and summation that the first terms on the right-hand side of (33) and (34) recover the two terms in (31) (assuming we have dropped $\sum_{\mathbf{x} \in \mathcal{X}_{[K]}} P_{[K]} \log \prod_{k \in [K]} \mu_k(x_k)$ from (31)). We may combine the second terms of (33) and (34) to obtain that the remaining term in the objective of (32) reads

$$\sum_{\mathbf{x} \in \mathcal{X}_{[T]}} p(\mathbf{x}) \int_{\mathcal{Y}_{[T]}} p(\mathbf{y}|\mathbf{x}) \log(\frac{r(\mathbf{y}|\mathbf{x})}{p(\mathbf{y}|\mathbf{x})}) d\mathbf{y}. \tag{35}$$

It follows from the strict concavity of $\log x$ that (35) is at most 0 and equals 0 if and only if $p(\mathbf{y}|\mathbf{x}) = r(\mathbf{y}|\mathbf{x})$ for every $\mathbf{x}$ and $p(\cdot|\mathbf{x})$-almost every $\mathbf{y}$. Since $r$ is a probability density function, the equality condition is equivalent to saying that for every $\mathbf{x} \in \mathcal{X}_{[T]}$ we have $p(\cdot|\mathbf{x}) = r(\cdot|\mathbf{x})$ Lebesgue almost everywhere. Therefore, if $P^*_{[T]}$ is a maximizer for (31) then $P^*_{[T]}(\mathbf{x}) r(\mathbf{y}|\mathbf{x})$ is a maximizer for (32). On the other hand, if $p^*(\mathbf{z})$ is a maximizer for (32), then $\int_{\mathcal{Y}_{[T]}} p^*(\mathbf{z}) d\mathbf{y}$ is a maximizer for (31).

$\square$

## C.3. Proof of Theorem 4.1

We first prove that if the initialization of Algorithm 1 and Algorithm 2 satisfy

$$\beta_k^{(0)} = v_k^{(0)} \quad \forall k \in [K] \tag{36}$$

then, at the end of each `while` loop in both algorithms,

$$\gamma_k^{(\ell)} = \mathbf{S}_k^{(\ell)},$$
$$\beta_k^{(\ell)} = v_k^{(\ell)}$$

for all $k \in [K]$.

We proceed by induction. We first investigate the first `for` loop in Algorithm 2 (the "left pass"). By backwards induction on $k = K, \ldots, 1$, we obtain that at the conclusion of the left pass

$$\delta_{k-1}^+(\mathbf{z}_{k-1}) = \int \cdots \int \sum_{x_k, \ldots, x_K} \prod_{i=k}^{K} \Phi_{i-1}(\mathbf{z}_{i-1}, \mathbf{z}_i) \beta_i^{(\ell-1)}(x_i) d\mathbf{y}_k \ldots d\mathbf{y}_K \quad \forall k \in [K]^+, \tag{37}$$

where the integration is taken over $(\mathbf{y}_k, \ldots, \mathbf{y}_K) \in \prod_{i=k}^{K} \mathcal{Y}_i$ and the sum is taken over $(x_k, \ldots, x_K) \in \prod_{i=k}^{K} \mathcal{X}_i$

Therefore, the updates to $\gamma_0$ and $\beta_0$ satisfy

$$\gamma_0^{(\ell)}(x_0) = \mathcal{I}_0(1, \delta_0^+)(\mathbf{z}_0)$$

$$= \sum_{x_1, \ldots, x_K} \int \cdots \int_{\mathcal{Y}_{[K]}} \prod_{i=1}^{K} \Phi_{i-1}(\mathbf{z}_{i-1}, \mathbf{z}_i) \beta_i^{(\ell-1)}(x_i) d\mathbf{y}$$

$$= \sum_{x_1, \ldots, x_K} \exp(-C(\mathbf{x})) \prod_{i=1}^{K} \beta_i^{(\ell-1)}(x_i)$$

$$= \mathbf{S}_0^{(\ell)}(x_0),$$

where the last equality uses the induction hypothesis $\beta_k^{(\ell-1)} = v_k^{(\ell-1)}$, and consequently

$$\beta_0^{(\ell)}(x_0) = \mu_0(x_0) / \mathbf{S}_0^{(\ell)}(x_0) = v_0^{(\ell)}(x_0).$$

We now show by induction on $k$ that $\gamma_k^{(\ell)} = \mathbf{S}_k^{(\ell)}$ and $\beta_k^{(\ell)} = v_k^{(\ell)}$ at the conclusion of the "right pass". We have already established the base case $k = 0$. For $k \geq 1$, as in the left pass, the updates in the right pass satisfy

$$\delta_k^-(\mathbf{z}_k) = \int \cdots \int \sum_{x_0,\ldots,x_{k-1}} \prod_{i=1}^k \Phi_{i-1}(\mathbf{z}_{i-1}, \mathbf{z}_i) \beta_{i-1}^{(\ell)}(x_i) d\mathbf{y}_0 \cdots d\mathbf{y}_k . \tag{38}$$

Combining (37) and (38) implies that

$$\begin{aligned}
\beta_k^{(\ell)}(x_k) &= \mathcal{I}_k(\delta_k^+, \delta_k^-) \\
&= \sum_{\substack{i=0 \\ i \neq k}}^K \int \cdots \int_{\mathcal{Y}_{[K]}} \prod_{i=0}^{K-1} \Phi_i(\mathbf{z}_i, \mathbf{z}_{i+1}) \prod_{i=0}^{k-1} \beta_i^{(\ell)}(x_i) \prod_{i=k+1}^K \beta_i^{(\ell-1)}(x_i) d\mathbf{y} \\
&= \sum_{\substack{i=0 \\ i \neq k}}^K \int \cdots \int_{\mathcal{Y}_{[K]}} \exp(-C(\mathbf{x})) \prod_{i=0}^{k-1} \beta_i^{(\ell)}(x_i) \prod_{i=k+1}^K \beta_i^{(\ell-1)}(x_i) \\
&= \mathbf{S}_k^{(\ell)}(x_k) ,
\end{aligned}$$

where the final equality holds by induction. As in the $k = 0$ case, this implies $\beta_k^{(\ell)} = v_k^{(\ell)}$. This proves the claim.

The first implication, on the time complexity of Algorithm 2, follows directly from existing analysis of multi-marginal entropic optimal transport problems (Altschuler & Boix-Adserà, 2023; Lin et al., 2022). Indeed, those works show that Algorithm 1 yields an $\varepsilon$-approximate solution to a discrete multi-marginal entropic optimal transport problem with arbitrary cost tensor $C$ in $\mathrm{Poly}(K, C_{\max}/\varepsilon)$ iterations. As each iteration of Algorithm 1 corresponds to an iteration of Algorithm 2, a similar guarantee holds for Algorithm 2. In particular, (Lin et al., 2022, Theorem 4.3 and proof of Theorem 4.5) shows that $K\bar{R}/\varepsilon$ iterations suffice, where $\bar{R}$ can be taken to be of order $C_{\max} + \log(KnC_{\max}/\varepsilon)$.

The second implication, on the form of the optimum, follows from the same considerations as Lemma 3.1. The proof of Lemma 3.1 establishes a direct link between solutions of (3) and (6). Specifically, if $P_{[T]}^*$ is the optimal solution for 3, then $p^*$ defined by $p^*(\mathbf{z}) := r(\mathbf{y}|\mathbf{x})P_{[T]}^*(\mathbf{x})$ is the corresponding solution for (6). Notice that $P^*(\mathbf{x}) \propto r(\mathbf{x}) \prod_{k=0}^K v_k^*(x_k)$, with $\{v_k^*\}$ being the fixed point of Algorithm 1. Thus,

$$p^*(z) \propto r(\mathbf{z}) \prod_{k=0}^K v_k^*(x_k) \propto \prod_{k=1}^K \Phi_k(\mathbf{z}_{k-1}, \mathbf{z}_k) \prod_{k=0}^K v_k^*(x_k),$$

proving (12). Since we have already shown that the iterations of Algorithms 1 and 2 agree, they have the same fixed points, and $v_k^* = \beta_k^*$, as desired.

$\square$

## C.4. Proof of Lemma 5.1

We compute:

$$\begin{aligned}
\ell_{k-1}^i(x_{k-1}) &= \int_{\mathcal{Y}_{k-1}} \phi_{k-1}^i(\mathbf{y}_{k-1}) \delta_{k-1}^+(\mathbf{z}_{k-1}) d\mathbf{y}_{k-1} \\
&= \sum_{x_k \in \mathcal{X}_k} \iint_{\mathcal{Y}_k \times \mathcal{Y}_{k-1}} \phi_{k-1}^i(\mathbf{y}_{k-1}) \Phi_k(\mathbf{z}_{k-1}, \mathbf{z}_k) \delta_k^+(\mathbf{z}_k) \beta_k(x_k) d\mathbf{y}_k d\mathbf{y}_{k-1} \\
&= \sum_{x_k \in \mathcal{X}_k} \sum_{j=1}^\infty \iint_{\mathcal{Y}_k \times \mathcal{Y}_{k-1}} \phi_{k-1}^i(\mathbf{y}_{k-1}) \Phi_k(\mathbf{z}_{k-1}, \mathbf{z}_k) \ell_k^i(x_k) \phi_k^j(\mathbf{y}_k) \beta_k(x_k) d\mathbf{y}_k d\mathbf{y}_{k-1} \\
&= \sum_{x_k \in \mathcal{X}_k} \beta_k(x_k) \sum_{j=1}^\infty \ell_k^j(x_k) \Gamma_k^{ij}(x_{k-1}, x_k) .
\end{aligned}$$

The computation for $r_k^i$ is analogous.

### C.5. Proof of Theorem 6.1

For each iteration of Algorithm 3, we run $2K$ sub-iterations for updating the left and right messages $l_k$ and $r_k$. The intermediate step (18) for the calculation of $\beta_k$ has complexity $O(nM)$. The major update steps (19) and (20) involve matrix-vector multiplication with a matrix of size $nM \times nM$ , which has complexity $O(n^2M^2)$. Therefore, the total complexity of the algorithm is given by $O(TKn^2M^2)$. $\qquad\square$

$\square$

## D. Practical implementation with the Haar basis

The Haar Wavelet basis stand out as an natural choice in the context of this work because of its positivity properties: since the messages need to be positive, using orthonormal bases that consist of positive functions guarantees that the approximation is positive as well. Recall that $\mathcal{X}_k$ is a finite collection with $n$ members and $\mathcal{Y}_k = \mathbb{R}^{m-1}$. Set a sequence of positive integers $M_1, \ldots, M_{m-1}$, with each number tied to a dimension of $\mathcal{Y}_k$. Our goal is to utilize $M = M_1 \times \cdots \times M_{m-1}$ orthonormal polynomials to execute the algorithm. Let $\vec{i} \in [M_1]^+ \times \cdots \times [M_{m-1}]^+$ with $\vec{i}_n$ representing the $n$-th component of $\vec{i}$. We define the following:

$$
\phi_k^{\vec{i}} := Z_k \mathbf{1}_{B_k^{\vec{i}}}
$$
$$
B_k^{\vec{i}} := \prod_{n=1}^{m-1} \left[ -\frac{K_n}{2}\delta_k + (\vec{i}_n - 1)\delta_k, -\frac{K_n}{2}\delta_k + \vec{i}_n\delta_k \right) \tag{39}
$$

Here, $Z_k$ serves as the normalization constant ensuring that $\phi_k^{\vec{i}}$ has a unit $L^2$-norm; effectively, $Z_k$ is the reciprocal of the volume of the hypercube $B_k^{\vec{i}}$. The parameter $\delta_k$ is chosen via:

$$
\frac{K_n}{2}\delta_k = C \cdot \sqrt{\max\left\{ \mathrm{Var}(\eta_k^{(2)}), \ldots, \mathrm{Var}(\eta_k^{(m)}) \right\}} \tag{40}
$$

where $C$ is a hyper-parameter that one can tune and a typical choice is 3. With the set $\{\phi_k^{\vec{i}}\}$ determined, we show how to compute $\boldsymbol{\Gamma}_0, ..., \boldsymbol{\Gamma}_{K-1}$ at the start of Algorithm 5. By setting $\delta_k$ to be very small, we can leverage this approximation:

$$
\Gamma_k(x_k, x_{k+1}, \vec{i}, \vec{j}) = \iint \phi_k^{\vec{i}}(\mathbf{y}_k)\phi_{k+1}^{\vec{j}}(\mathbf{y}_{k+1})\Phi_k(\mathbf{z}_k, \mathbf{z}_{k+1})d\mathbf{y}_k d\mathbf{y}_{k+1} \tag{41}
$$
$$
\approx \mathrm{Vol}(B_k^{\vec{i}})\mathrm{Vol}(B_{k+1}^{\vec{j}})\Phi_k(x_k, \mathbf{y}_k^{\vec{i}}, x_{k+1}\mathbf{y}_{k+1}^{\vec{j}}), \tag{42}
$$

where $\mathbf{y}_k^{\vec{i}}$ is the midpoint of the hypercube $B_k^{\vec{i}}$. Remember, $\Phi_k(\mathbf{z}_k, \mathbf{z}_{k+1})$ indicates the conditional density of the reference process $r(\mathbf{z}_{k+1}|\mathbf{z}_k)$ for $k \geq 1$, and shifts to the unconditional joint density $r(\mathbf{z}_0, \mathbf{z}_1)$ at $k = 0$. Evaluating the relevant Gaussian densities allows us to compute $\Gamma_k(x_k, x_{k+1}, \vec{i}, \vec{j})$ efficiently.

When Algorithm 5 reaches convergence, the representation in (12) can be used to efficiently manipulate $p$ (for instance, to generate samples). In particular, (12) shows that $p$ inherits the Markov property of $r$, so to sample from $p$ it suffices to compute the pairwise marginals $p(\mathbf{z}_k, \mathbf{z}_{k+1})$, which are proportional to

$$
\int \cdots \int \sum_{\substack{i \in [K] \\ i \neq k, k+1}} \sum_{x_i \in \mathcal{X}_k} \prod_{k=1}^{K} \Phi_k(\mathbf{z}_{k-1}, \mathbf{z}_k) \prod_{k=0}^{K} \beta_k^*(x_k)d\mathbf{y}_0 \cdots d\mathbf{y}_{k-1}d\mathbf{y}_{k+2} \cdots \mathbf{y}_K .
$$

This may be computed efficiently by repeatedly contracting one coordinate at a time, as in the left pass of Algorithm 2.

When Algorithm 3 is implemented using wavelets, $p(\mathbf{z}_k, \mathbf{z}_{k+1})$ can be approximated by $\mathbf{p}_k(x_k, \vec{i}, x_{k+1}, \vec{j})$, which approximates $p(x_k, y_k^{\vec{i}}, x_{k+1}, y_{k+1}^{\vec{j}})$. With $\mathbf{p}(x_k, \vec{i}, x_{k+1}, \vec{j})$ in hand, we propose two trajectory inference methods. The first method, which is stochastic, generates trajectory samples following the distribution $p(\mathbf{z}_k, \mathbf{z}_{k+1})$, as outlined in Algorithm 4.

The other way to rebuild the trajectories involves utilizing argmax operations on the belief tensor, and it is summarized as Algorithm 5

---

**Algorithm 4** Trajectory Inference Through Sampling

> **Input:** $\mathbf{p}_0, ..., \mathbf{p}_{K-1} \in \mathbb{R}^{n \times n \times M \times M}$ and $x_0 \in \mathcal{X}_0$
> Marginalize $\mathbf{p}_0$ to a probability distribution on $\mathcal{X}_0 \times \{\mathbf{y}_0^{\vec{i}}\}$, denote it as $\mathbf{p}_{\text{int}}$
> Sample $\mathbf{y}_0$ proportional to $\mathbf{p}_{\text{int}}(x_0, \cdot)$
> **for** $k = 0, ..., K-1$ **do**
>    Let $\vec{i}_k$ be the index of $\mathbf{y}_k$ in $\{\mathbf{y}_k^{\vec{i}}\}$
>    Sample $(x_{k+1}, \mathbf{y}_{k+1})$ proportional to $\mathbf{p}_k(x_k, \vec{i}_k, \cdot, \cdot)$
> **end for**
> Return $\{x_0, ..., x_K\}$ and $\{y_0, ..., y_K\}$

---

**Algorithm 5** Trajectory Inference Through ArgMax

> **Input:** $\mathbf{p}_0, ..., \mathbf{p}_{K-1} \in \mathbb{R}^{n \times n \times M \times M}$ and $x_0 \in \mathcal{X}_0$
> Marginalize $\mathbf{p}_0$ to a probability distribution on $\mathcal{X}_0 \times \{\mathbf{y}_0^{\vec{i}}\}$, denote it as $\mathbf{p}_{\text{int}}$
> Set $\mathbf{y}_0$ as the maximizer of $\mathbf{p}_{\text{int}}(x_0, \cdot)$
> **for** $k = 0, ..., K-1$ **do**
>    Let $\vec{i}_k$ be the index of $\mathbf{y}_k$ in $\{\mathbf{y}_k^{\vec{i}}\}$
>    Set $(x_{k+1}, \mathbf{y}_{k+1})$ as the maximizer of $\mathbf{p}_k(x_k, \vec{i}_k, \cdot, \cdot)$
> **end for**
> Return $\{x_0, ..., x_K\}$ and $\{y_0, ..., y_K\}$

---

## E. Log-domain Implementation of Algorithm 5

In the execution of the algorithm, some values could fall below machine precision, triggering numerical problems. Take, for instance, the wavelet basis: we aim to estimate $\Gamma_k$ by computing $\Phi_k$ at a specific point. However, if $\mathbf{y}_k$ or $\mathbf{y}_{k+1}$ drifts significantly away from its mean, $\Phi_k$ ends up zero. For the sake of numerical stability, we derive the log-domain implementation of Algorithm 5 in the section. Recall that $M$ is the total number of orthonormal polynomials that we use to perform the approximate message passing algorithm. Define the log version of quantities

$$\hat{\ell}_k(x_k) := \log(\ell_k(x_k)) \text{ and } \hat{r}_k(x_k) := \log(r_k(x_k)) \tag{43}$$

$$\hat{c}_k(x_k) := \log(c_k(x_k)) \text{ and } \hat{\Gamma}_k^{i,j}(x_k, x_{k+1}) := \log\left(\Gamma_k(x_k, x_{k+1}, i, j)\right) \tag{44}$$

For $k = 0, ..., K$, the log-domain updates are summarized as follows:

$$\hat{\ell}_k^i(x_k) = \log\left(\sum_{x_{k+1} \in \mathcal{X}_{k+1}} \sum_{j=1}^M \exp\left(\hat{c}_{k+1}(x_{k+1}) + \hat{\ell}_{k+1}^j(x_{k+1}) + \hat{\Gamma}_k^{i,j}(x_k, x_{k+1})\right)\right) \tag{45}$$

$$\hat{r}_t^i(x_t) = \log\left(\sum_{x_{t-1} \in \mathcal{X}_{t-1}} \sum_{j=1}^M \exp\left(\hat{c}_{t-1}(x_{t-1}) + \hat{r}_{t-1}^j(x_{t-1}) + \hat{\Gamma}_{t-1}^{j,i}(x_{t-1}, x_t)\right)\right) \tag{46}$$

$$\hat{c}_k(x_k) := \log(c_k(x_k)) = -\log\left(\sum_{j=1}^M \exp\left(\hat{\mathbf{l}}_k^j(x_k) + \hat{\mathbf{r}}_k^j(x_k)\right)\right).$$

Note that one can use the `logsumexp` function in scientific computing packages to implement the log-domain updates given above. And, for the wavelet decomposition, $\hat{\Gamma}_k^{i,j}(x_k, x_{k+1})$ can be efficiently approximated by calling a SciPy `scipy.stats.norm.logpdf` function.

## F. Experiment Details

We first summarize the different data sets, metrics, and experimental findings.

For all experiments, we consider datasets with the number of points being constant at each step and each particle has uniform weight. For the kernel, we use either Matern kernel with $\nu = 1.5$ or $\nu = 2.5$.

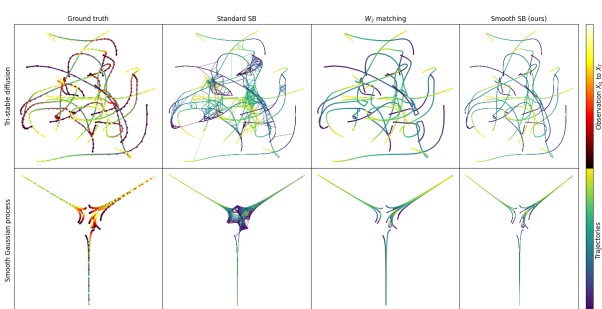

*Figure 7.* Visualization of matching results on Tri-stable diffusion and 2D Matern datasets.

### F.1. Individual particle tracking

**Smooth Gaussian process: (2D version of Figure 2)**: Each particle follows a $2D$ Gaussian process with dimension-independent covariance matrix generated by Matern kernel with $\nu = 3.5, \ell = 1, \sigma = 1$. Our algorithm can recover most of the trajectories correctly by selecting smoother trajectories than the other two algorithms.

**Tri-stable diffusion process:** Particles are initialized randomly around zero by a normal distribution and follow the evolution of a deterministic dynamic system. This dynamic is considered in (Lavenant et al., 2024), each particle will be absorbed into one of the three attraction points with increasing velocity as the distance to the attraction point decreases. We find when particles are close to each other and have similar velocities, our algorithm has the chance to lose track of the particles during sampling. The $W_2$ matching performs almost perfectly in this scenario by choosing the closest particle.

**N-body physical system:** This dataset simulates the orbits of eight planets that circle a star and the task is to keep track of each planet's orbit in the system. The 2D version is considered in (Chewi et al., 2021), but they only consider smooth interpolations between objects and provide no guarantee of the correctness of matching. The visualization result is presented in Figure 4. The $W_2$ matching and standard SB make many mistakes when trajectories are close to each other while our algorithm gives near-perfect tracking.

Our quantitative evaluation appears in Table 1. We use several metrics to evaluate our results: $\mathrm{JumpP}$ is the observed likelihood that a particle transitions to a different path at each time increment. $\mathrm{5p\ acc}$ quantifies the proportion of a sequence of five consecutive steps that remain on the same trajectory. The $\mathrm{Mean}\ \ell_2$ metric is the average of all $\ell_2$ distances between each sampled trajectory and the ground truth trajectory that maintains the longest continuous alignment with the sample.

### F.2. Point cloud inference

**Petal:** The Petal dataset was introduced in (Huguet et al., 2022b) which mimics natural dynamics of cellular differentiation and bifurcation.

**Converging Gaussian:** This dataset was constructed by (Clancy & Suarez, 2022). It models cellular dynamics using Gaussian point clouds that split and converge over time. Points are sampled evenly around the center of the circles.

**Embryoid Body:** This sc-RNAseq dataset records statistical data of embryoid body (EB) differentiation over a period of 27 days with 5 snapshots taken between days of 0-3, 6-9, 12-15, 18-21, and 24-27. The dimension of the original data was reduced to 2 using a nonlinear dimensionality reduction technique called PHATE (Moon et al., 2019) in (Tong et al., 2020). We inherit this 2D projection dataset from them.

**Dyngen Tree (Dyg Tree):** This is another sc-RNAseq dataset crafted by (Huguet et al., 2022b) using Dyngen (Cannoodt et al., 2021). The data is embedded into dimension 5 also using PHATE. This dataset contains one bifurcation and is considered to be more challenging than the Petal dataset.

**Dyngen Cycle (Dyg Cycle):** Another 10-D sc-RNAseq dataset inherited from (Banerjee et al.).

We subsample each dataset to a moderate size and make $n_t$ constant over all timesteps.

Our quantitative evaluation uses the following metrics: $\mathbf{W}_1$ is the Wasserstein-$\ell_1$ distance, $\mathbf{M}_G$ is the Maximum Mean Discrepancy with Gaussian kernel $K(x,y) := \sum_{i,j=1}^{n}(x_i - y_j)^2/2$, and $\mathbf{M}_I$ is the Maximum Mean Discrepancy with identity kernel $K(x,y) = \langle x, y \rangle$.

Table 3 presents additional results on other LOT tasks.

*Table 3.* Performance comparison on LOT tasks at step $j$ between our algorithm and the state-of-art algorithms.

| Dataset | Method | $\mathbf{W}_1(\downarrow)$ | $\mathbf{M}_G(\downarrow)$ | $\mathbf{M}_I(\downarrow)$ |
|---|---|---|---|---|
| Petal $j=4$ | SSB (ours) | *2.98e-2* | **3.26e-5** | **5.53e-3** |
| | MIOFlow | 1.76e-1 | 9.97e-3 | 1.14e-1 |
| | DMSB | 2.63e-1 | 9.18e-3 | 1.15e-2 |
| | F&S | **2.44e-2** | *4.41e-5* | *7.33e-3* |
| EB $j=3$ | SSB (ours) | **6.00e-2** | **3.88e-4** | **2.04e-2** |
| | MIOFlow | 1.29e-1 | 3.31e-3 | 5.26e-2 |
| | DMSB | 2.46e-1 | 4.69e-2 | 2.37e-1 |
| | F&S | *7.42e-2* | *1.46e-3* | *3.88e-2* |
| Dyg Tree $j=2$ | SSB (ours) | *1.09e-1* | *4.67e-3* | *6.73e-2* |
| | MIOFlow | 2.23e-1 | 1.57e-2 | 1.34e-1 |
| | DMSB | * | * | * |
| | F&S | **9.23e-2** | **1.43e-3** | **3.93e-2** |

## F.3. Implementation details

We ran our experiments on an x86-64 setup. Smooth SB (ours) and F&S do not require GPU support; for DMSB and MIOFlow experiments, we utilized an NVIDIA A100-SXM4-80GB. In tracking individual particles, standard SB with Brownian motion prior and $\mathbf{W}_2$ matching is implemented by Python's OT library (Flamary et al., 2021). Check out Table 4 for detailed dataset info.

*Table 4.* The table provides detailed statistics of all the datasets we experimented on. Since the number of particles is constant over time, we use $n$ to denote the number of observations at each time step and $K$ stands for the total number of time steps excluding the initial observation. The Figure column stands for the label of figures where this dataset is presented in the main body text.

| Name | $K$ | $n$ | Dimension | Figure | Author |
|---|---|---|---|---|---|
| Smooth Gaussian Process | 20 | 20 | 1 | 2 | us |
| Smooth Gaussian Process (2D) | 20 | 25 | 2 | 7 | us |
| Tri-stable Diffusion | 20 | 20 | 2 | 7 | (Lavenant et al., 2024) |
| N Body | 50 | 8 | 3 | 4 | us |
| Petal | 5 | 40 | 2 | 1,5 | (Huguet et al., 2022b) |
| Converging Gaussian | 3 | 48 | 2 | 5 | (Clancy & Suarez, 2022) |
| Embryoid Body | 4 | 100 | 2 | 5 | (Tong et al., 2020) |
| Dyngen Tree | 5 | 40 | 5 | 5 | (Huguet et al., 2022b) |
| Dyngen Cycle | 15 | 25 | 10 | 6 | (Banerjee et al.) |

Throughout our experiments, we employ either a Matern prior with parameters $\nu = 1.5, \ell = 3$ or $\nu = 2.5, \ell = 2$, as specified in (5), utilizing the wavelet basis. We conduct 200 iterations of message-passing algorithms ($T = 200$). The initial covariance, $\tilde{\Sigma}_{0,0}$, is derived from the stationary distribution's covariance of the lifted Gaussian process corresponding to

the kernel, ensuring $\tilde{\Sigma}_{k,k}$ remains steady over time (refer to Equation (2.39) in (Saatçi, 2012) for stationary distribution calculation). We assume all observations occur equally over the time period $t = 2$, hence the observation duration is $dt = 2/K$, which is utilized to compute covariance between timesteps. For datasets with dimensions greater than one, the kernel is assumed to be identical and independent for each dimension, simplifying the tensor $\Gamma$ by factoring it across dimensions, thereby reducing the computational load in the matrix-vector multiplication during the message-passing phase. We assume an equal allocation of coefficients per dimension, thus $M_i = (M)^{\frac{1}{d}}$ for each $i = 1, ..., d$. For the kernel with $\nu = 2.5$, we express $M_i = (m_1, m_2)$ as a tuple where $M_i = m_1 m_2$ represents the count of approximation coefficients in the velocity and acceleration dimension. Each task has a fixed $M_i$ choice except for the Dyngen Cycle dataset, with the kernel's variance calculated as $\sigma = c\sigma_{data}$, where $\sigma_{data}$ is the dataset's standard deviation and $c$ is an adjustable hyper-parameter. Detailed parameter settings for each task can be found in 5 and 6.

*Table 5.* Detailed parameter setting of our algorithm for all the experiments for Figures 2, 7, 5 and 4

| Dataset Name | $\nu$ | M | $M_i$ | Dimension | $c$ |
|---|---|---|---|---|---|
| Smooth Gaussian Process | 2.5 | 200 | (40,5) | 1 | 1 |
| Smooth Gaussian Process (2D) | 2.5 | 1024 | (8,4) | 2 | 1 |
| Tri-stable Diffusion | 1.5 | 1024 | 32 | 2 | 1 |
| N Body | 2.5 | 1728 | (4,3) | 3 | 4 |
| Petal | 1.5 | 400 | 20 | 2 | 0.5 |
| Converging Gaussian | 1.5 | 900 | 30 | 2 | 0.5 |
| Embryoid Body | 1.5 | 144 | 12 | 2 | 1 |
| Dyngen Tree | 1.5 | 1024 | 4 | 5 | 0.5 |
| Dyngen Cycle | 1.5 | 1024 | $(4) \times 5 + (1) \times 5$ | 10 | 2.5 |

*Table 6.* Detailed parameter setting of our algorithm for the LOT tasks in Table 3 and 2 where we only used Matern kernel with $\nu = 1.5$ and all the parameters are the same as the corresponding experiment in Table 5 except for $c$.

| | | Petal $j = 2$ | Petal $j = 4$ | EB $j = 2$ | EB $j = 3$ | Dyg Tree $j = 1$ | Dyg Tree $j = 2$ | Dyg Cycle $j = 7$ |
|---|---|---|---|---|---|---|---|---|
| $c$ | | 0.5 | 0.45 | 1 | 1 | 0.3 | 0.25 | 2.5 |

**Implementation of Sampling:** In the point cloud matching, where creating trajectories or inferring positions of the left-out timestep requires generating new points, we use conditional Gaussian sampling. Initially, $\mathbf{y}_k$ for all data points are sampled from calculated probability tensors $\mathbf{p}_k$. Under wavelet basis, the sample we get pins down the range of the $\mathbf{y}_k$, we then calculate the conditional Gaussian density for $\mathbf{y}_k$ given $\mathbf{x}_k$ and simply use sampling and rejecting to get a valid velocity sample. Given that the lifted Gaussian process is Markovian, we condition on two consecutive observations to generate extra points for constructing trajectories or estimating positions at unobserved timesteps.

## F.4. Details of other algorithms

**Standard Schrödinger Bridges:** The single hyper-parameter in regular SB is the scaling factor $s$ for the correlation between successive time steps, defined as $\text{cov}(i, j) = s \min(i, j)$. Table 7 shows the configuration.

*Table 7.* Detailed parameter setting of standard Schrödinger Bridges

| | Smooth Gaussian process (1D & 2D) | N-Body | Tri-stable Diffusion |
|---|---|---|---|
| $s$ | 0.5 | 0.3 | 0.03 |

**F&S:** Natural cubic splines were our interpolation method of choice, as detailed in Chewi et al., 2021, Appendix D, and we wrote our own code for implementation. The visual results of the same experiments in Figure 5 are shown in Figure 8 & 9. In the case of the Dyngen dataset, F&S's matching is significantly poorer than what our algorithm achieves. This underscores a crucial flaw of the standard $W_2$ matching—it lacks the capacity to split mass effectively when the data distribution is uniform. The primary culprit is Dyngen's asymmetric bifurcation structure.

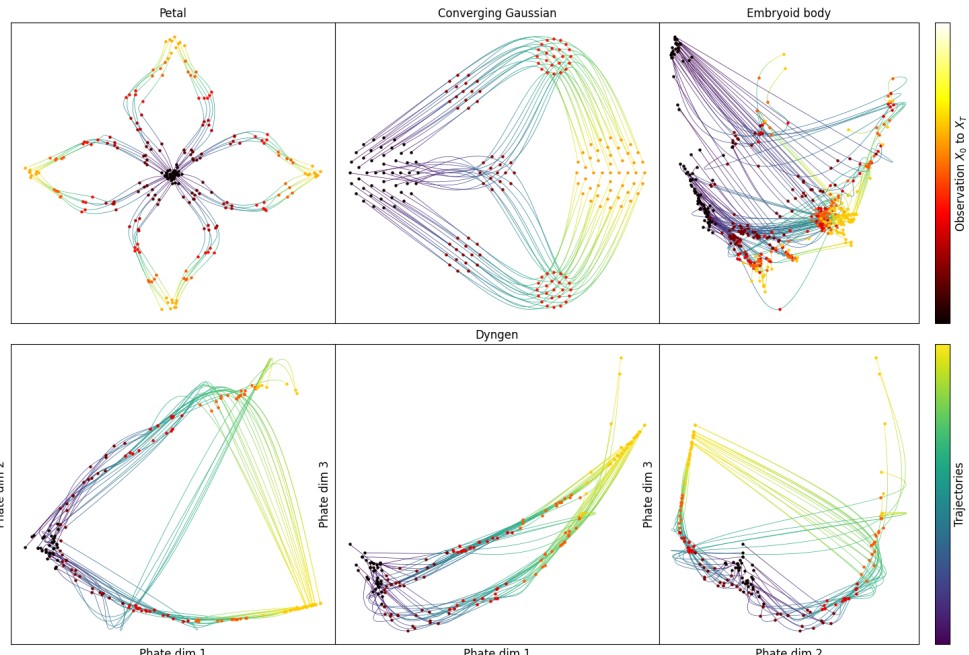

*Figure 8.* Visualization of trajectory inference on various dataset by F&S, for the 5D Dyngen Tree data, we visualize the 2D projection of the first three dimensions in the second row.

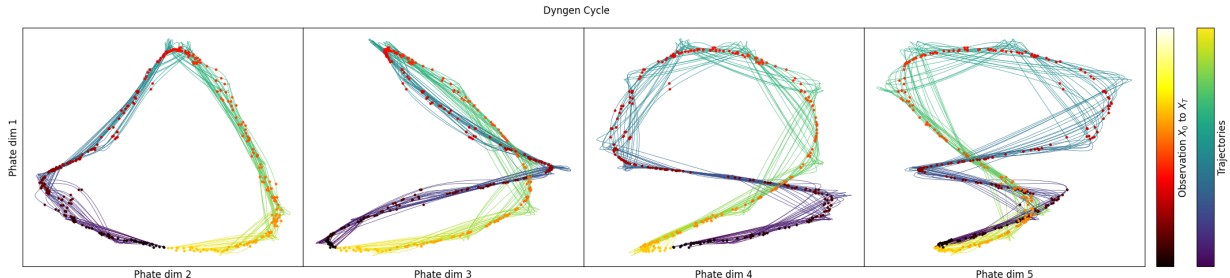

*Figure 9.* Visualization of trajectory inference on various dataset by FSI, for the 10D Dyngen Cycle data, we visualize the 2D projection of the first dimension against the second to fifth dimensions.

**MIO Flow** [2]**:** We build upon the baseline settings from (Huguet et al., 2022b), with tuning for optimal performance. Here's the prime parameter configuration we achieved:,

1. **Petal (complete):** $\lambda_e = 1e - 2, \lambda_d = 25$, other variables adhere to the default Petal ($hold\_one\_out = False$) setup.

2. **Petal (leave one out):** $\lambda_e = 1e - 3, \text{n\_local\_epochs} = 40, \text{n\_epochs} = 0$, remaining parameters align with the default Petal ($hold\_one\_out = True$) configuration.

3. **Embryoid Body (complete & leave one out):** $\text{gae\_embeded\_dim} = 2$, other settings match the default Embryoid Body configuration.

---

[2]https://github.com/KrishnaswamyLab/MIOFlow

4. **Converging Gaussian (complete):** $\lambda_e = 1e - 3, \lambda_d = 15, \text{n\_local\_epochs} = 20$, all others follow the default Petal (hold_one_out = False) guidelines.

5. **Dyg Tree & Cycle (complete):** parameters match the default Dyngen (hold_one_out = False) specifications.

6. **Dyg Tree & Cycle (leave one out):** $\text{n\_local\_epochs} = 0, \text{n\_epochs} = 50$, rest conforms to the default Dyngen (hold_one_out = True) settings.

,Visual comparisons for these experiments, as seen in Figure 5, are presented in Figure 10 & 11. **DMSB**[3]**:** By primarily

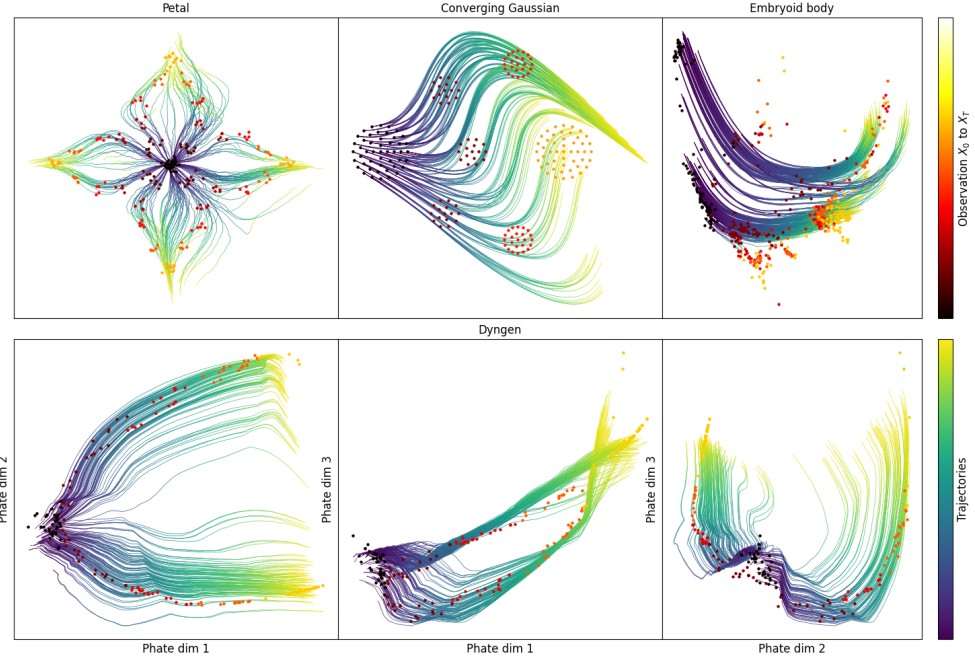

*Figure 10.* Visualization of trajectory inference on various dataset by MIO, for the 5D Dyngen Tree data, we visualize the 2D projection of the first three dimensions in the second row.

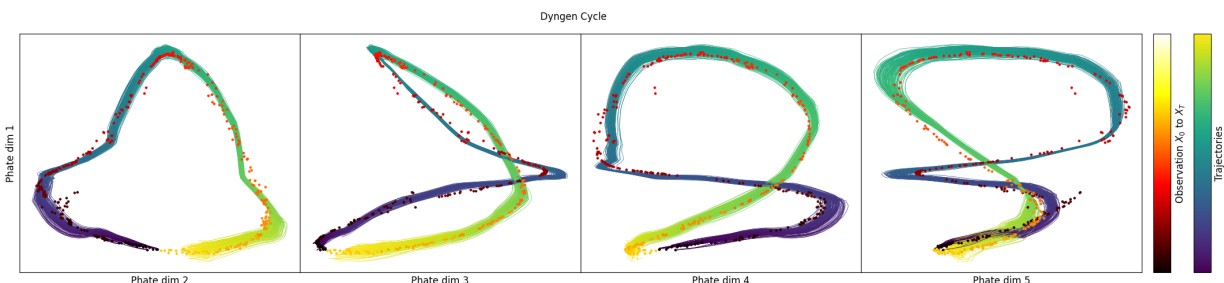

*Figure 11.* Visualization of trajectory inference on various dataset by MIO, for the 10D Dyngen Cycle data, we visualize the 2D projection of the first dimension against the second to fifth dimensions.

using the default parameters from (Chen et al., 2023), alongside some refinements, here's the top-notch parameter set:,

1. **Petal (leave one out):** $\text{n\_epoch} = 2, \text{num\_stage} = 13, num\_marg = 6$, other values are consistent with the default Petal setup.

---

[3]https://github.com/TianrongChen/DMSB

2. **Embryoid Body (leave one out):** $n\_epoch = 2, num\_stage = 13$, other settings align with the default Petal configuration.

, tuning this algorithm as each network requires approximately 5 hours for training. We anticipate enhanced DMSB performance with more extensive data and training duration. Moreover, the trajectory generation process of DMSB does not capture the geometry of the dataset well (see Figure 12 as an illustration), hence we do not provide additional visualization for this algorithm here.

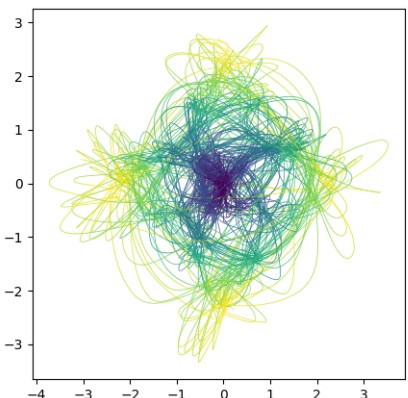

*Figure 12.* Trajectories drawn by DMSB can not reflect the underlying geometric structure of the Petal dataset well.

### F.5. Additional Results and Analysis

**Extended quantitative results on one-by-one particle tracking:** We provide more metrics in Table 8 for evaluating the performance of one-by-one particle tracking, which is an extend version of Table 1.

- JumpP: the observed likelihood that a particle transitions to a different path at every increment in time.
- 3p acc: the proportion of a sequence of three consecutive steps that remain on the same trajectory.
- 5p acc: the proportion of a sequence of five consecutive steps that remain on the same trajectory.
- Traj acc: the fraction of trajectories sampled that are correctly matched at each step.

For a given sampled trajectory, we match it with a ground-truth trajectory that maintains the longest continuous alignment with the sample. We say this ground truth trajectory is the *matched* trajectory of the sample.

- Max $\ell_2$: the maximum $\ell_2$ distance found between a sampled trajectory and its matched ground truth across every sample.
- Mean $\ell_2$: the average $\ell_2$ distance found between a sampled trajectory and its matched ground truth across every sample.
- TrajKL: the KL-divergence between the matched trajectory histogram and the uniform distribution over the ground-truth trajectories.

**Visualization of LOT tasks:** We provide the visualization of the sampled point in the Leave-One-Out tasks in Figures 13 and 15 corresponding to results in Table 2. For all the cases, our algorithm is able to draw similar patterns to the ground truth.

**Impact of scale hyper-parameter $c$:** Recall that the variance of the Matérn kernel in our experiments is chosen to be $\sigma = c\sigma_{\mathrm{data}}$ for a positive hyperparameter $c$. We need to choose an appropriate $c$ such that the algorithm is able to search for

| Dataset | Model | JumpP | 3p acc | 5p acc | Traj acc | Max $\ell_2$ | Mean $\ell_2$ | TrajKL |
|---------|-------|-------|--------|--------|----------|--------------|---------------|--------|
| 1D Matérn Process | SSB (ours) | **4.77e-2** | **0.930** | **0.893** | **0.530** | **1.912e-01** | **2.552e-03** | **0.000360** |
| | BME | 6.08e-1 | 0.191 | 0.079 | 0.001 | 1.463e+00 | 2.708e-01 | 0.036132 |
| | W2 | 2.18e-1 | 0.621 | 0.412 | 0.050 | 4.641e-01 | 1.526e-01 | 0.138629 |
| Tri-stable Diffusion | SSB (ours) | 1.12e-1 | 0.852 | 0.798 | 0.489 | 1.020e-01 | 1.419e-02 | 0.011770 |
| | BME | 5.20e-1 | 0.300 | 0.170 | 0.018 | 6.787e-01 | 6.595e-02 | 0.047383 |
| | W2 | **2.25e-2** | **0.971** | **0.956** | **0.800** | **1.377e-08** | **2.112e-09** | **0.000000** |
| N body | SSB (ours) | **5.00e-4** | **0.999** | **0.999** | **0.983** | 2.083e-01 | **5.475e-04** | **0.000100** |
| | BME | 1.14e-1 | 0.796 | 0.641 | 0.000 | 9.719e-01 | 5.426e-01 | 0.026030 |
| | W2 | 1.08e-1 | 0.804 | 0.649 | 0.000 | 8.679e-01 | 5.495e-01 | 0.173287 |
| 2D Matérn Process | SSB (ours) | **5.60e-3** | **0.993** | **0.991** | **0.904** | **4.181e-03** | **3.571e-04** | **0.000000** |
| | BME | 1.38e-1 | 0.764 | 0.612 | 0.179 | 1.418e+00 | 2.819e-01 | 0.038109 |
| | W2 | 6.60e-2 | 0.867 | 0.760 | 0.360 | 1.015e+00 | 2.321557e-01 | 0.110904 |

*Table 8.* Performance metrics for different models across various datasets for particle matching.

the next matching point that constructs a smooth interpolation. If $c$ is too small, then the velocity space of the algorithm searching will not be enough to incorporate the correct particles, this leads to the consequence that some of the particles will not be matched at all. If $c$ is too large, then it will result in more randomness in the algorithm, which will hurt the performance of OBO tracking and blur the underlying geometric structure in pattern matching. See Figure 16 on the Petal dataset for illustrations. In practical implementation, we increase $c$ if we find all the sampled velocities are concentrated at the center of the velocity grids and decrease $c$ if they are concentrated at the border instead.

**Time and Space Complexity Analysis:** our algorithm spends most of the time running the message passing algorithm. The running time complexity of one iteration of message passing is given by $O(KM^2n^2)$. Since we use dimension independent kernel, the tensor can be factorized across dimensions, we take $M_i = M^{\frac{1}{d}}$, and a single iteration in this case has complexity $O(dTM^{1+\frac{1}{d}}n^2)$, this allows us to take more coefficient when dimension increases. We give the run time record for one iteration of message passing when $T = 10$ and $N = 20$ in Table 9

*Table 9.* Running time record (in seconds) of one iteration of message passing, $M_i$ is taken to be $round(M^{1/d})$. One observes that the running time decreases for fixed $M$ when $d$ increases due to the dimension independence.

| | d=1 | d=2 | d=3 |
|---|-----|-----|-----|
| M=1024 | 80.91 | 6.32 | 5.21 |
| M=512 | 20.87 | 2.59 | 1.29 |
| M=256 | 5.36 | 0.82 | 0.45 |
| M=128 | 1.42 | 0.26 | 0.26 |

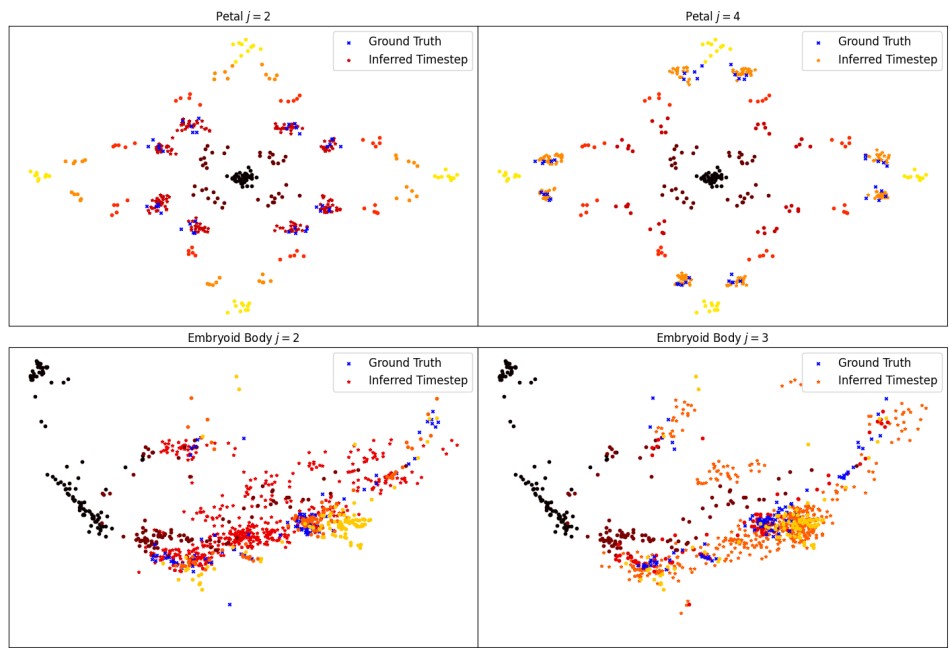

*Figure 13.* Visualization of sampled timestep in the LOT tasks for Dyngen datasets. The blue 'x' points are the left-out ground truth, the star points are inferred timestep by our algorithm.

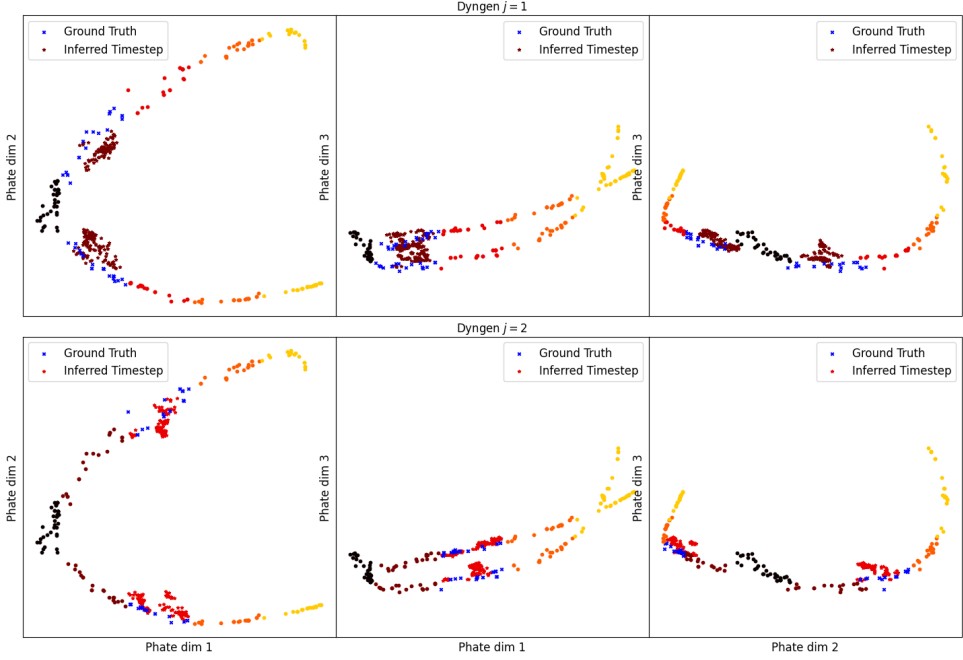

*Figure 14.* Visualization of sampled timestep in the LOT tasks for Dyngen Tree dataset. The blue 'x' points are the left-out ground truth, the star points are inferred timestep by our algorithm.

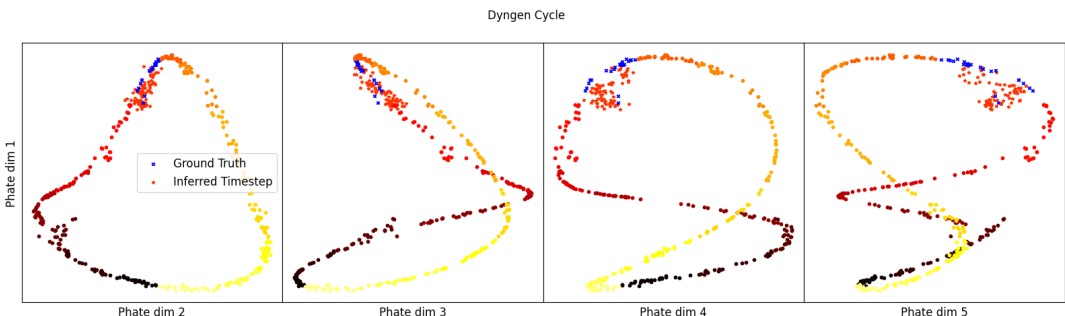

*Figure 15.* Visualization of sampled timestep in the LOT tasks for Dyngen Cycle datasets. The blue 'x' points are the left-out ground truth, the star points are inferred timestep by our algorithm.



*Figure 16.* Visualization of matching on the Petal dataset. We observe when $c = 0.1$ some observations have no matching at all, while when $c = 1, 2$, the matching contains more randomness and mistakes.

