# OpenReview forum: "Trajectory Inference with Smooth Schrödinger Bridges"
_ICML.cc/2025/Conference — ICML 2025 poster_

### Official Review · Reviewer_1euq · 2025-02-20

**Overall Recommendation:** 2

**Summary:**

The authors proposed the Schrodinger Bridge (SB) with smooth priors guided by the Gaussian process to reduce the exponential cost in K for solving multi-marginal SB problems. The problem is first discretized (suffers from exponential cost) and then lifted to high dimensions and solved by belief propagation methods, resulting in polynomial costs.

**Claims And Evidence:**

The paper is a bit hard to read.

The multi-marginal SB problem is
1) first discretized into discrete space in sec 2;
2) augmented with a new continuous variable and was related to a sequential conditional structure via the Gauss-Markov property;
3) was solved using a linear Belief Propagation (BP);
4) a practical (BP).

Overall, I think the method is quite hard to understand, especially when I am not an expert in the Gaussian process and have no knowledge in  belief propagation.

**Essential References Not Discussed:**

NA

**Experimental Designs Or Analyses:**

I don't know much about the usage of multi-marginal SB or the applications in RNA-Seq.

**Methods And Evaluation Criteria:**

NA

**Other Comments Or Suggestions:**

The first paragraph in section 3 is quite important. I suggest the authors make a diagram of the flow of methods to facilitate the reading.

**Other Strengths And Weaknesses:**

I am a bit confused by the usage of first considering the discrete state space and then uplifting the problem to a higher dimension and even augmenting with some continuous variables.

It seems like we don't need the discretization step. As such, the exponential cost does not exist actually when we don't consider the discrete-type problems.

**Questions For Authors:**

1) limitation: its trajectories inherit the roughness of Brownian paths, leading to noisier estimators and less interpretable posterior paths. This problem can be easily solved by the probability flow ODE, why cannot we use the prob flow ODE to avoid the Brownian motion?

2) Can we tackle the continuous multi-marginal SB directly without tackling the discrete optimal transport? So many tools have been used, which may affect the adoption of this method.

**Relation To Broader Scientific Literature:**

This paper aims to alleviate the exponential cost when it comes to many marginals.

**Theoretical Claims:**

Didn't check.

---

> ### Author Rebuttal · Authors · 2025-03-31
>
> We thank the reviewer for these  comments.
>
> **Comment:**
>
> Can we use the probability flow ODE to avoid Brownian motion?
>
> **Response:**
>
> While probability flow ODEs are indeed powerful tools, they are not ideal for our specific goals for several reasons:
>
> 1. Inferring Individual Trajectories:
>     - Probability flow ODEs focus on evolving the entire distribution and don't preserve individual particle identities or trajectories
>     - Our method explicitly tracks individual particles, which is crucial for understanding the motion of individual particles (e.g., in N-body systems)
>
> 2. Statistical Framework:
>     - The smooth Schrödinger bridge has an elegant quasi-Bayesian interpretation: the Schrödinger bridge matches the observed marginal distributions while keeping particle trajectories as faithful as possible to the prior $R$
>     - Probability flow ODEs lack such statistical guarantees when used for smoothing
>
> 3. Other Advantages of our Method:
>     - Our method naturally provides velocity and acceleration estimates. These emerge directly from the HMM model we use
>     - Post-hoc smoothing via ODEs would require additional assumptions and computation
>
> This makes our approach more suitable for applications requiring detailed trajectory analysis and closer adherence to the prior distribution.
>
> **Comment:**
>
> Can we tackle the continuous multi-marginal SB directly without tackling the discrete optimal transport?
>
> **Response:**
>
> The goal of this paper is to infer particle trajectories that are as close to the prior path distribution as possible while matching the observed empirical distributions. Empirical distributions are discrete by nature, so a discrete multi-marginal Schrödinger bridge problem is a natural starting point. However, our algorithm can be extended to continuous marginals (population distribution) by using the following approach:
> 1. Replace the messages $\beta$ and $\gamma$ with their continuous versions
> 2. Replace sums with integrals in (8) on the right-hand side of the equation
>
> This continuous extension of our algorithm yields scaling functions $\beta^*_0, ..., \beta^*_K$ that correspond to the multi-marginal Sinkhorn algorithm for continuous distributions. Crucially, this modification preserves the computational efficiency of the discrete version, reducing the exponential complexity of multi-marginal Sinkhorn to polynomial time. Furthermore, similar to how we represent $\delta$'s using orthonormal bases in Section 5, the continuous $\beta$ functions can also be expressed through basis expansion, maintaining both mathematical rigor and computational tractability.
>
> **Comment:**
>
> I suggest the authors make a diagram of the flow of methods to facilitate the reading.
>
> **Response:**
> We appreciate this suggestion and will add a diagram to the revised version of the paper. The diagram will illustrate the flow of methods, including the relationship between the multi-marginal Schrödinger bridge problem, the static problem, and the lifted space. This will help clarify the overall structure and flow of our approach for readers.

---

### Official Review · Reviewer_okGt · 2025-02-26

**Overall Recommendation:** 3

**Summary:**

**Disclaimer: Despite the forthcoming criticisms, I find this paper intriguing and recommend its acceptance.**


------------------------------

This paper introduces a class of smooth Gaussian processes as priors for Schrödinger bridges and designs efficient algorithms for their computation. A key insight is that, while these processes are inherently non-Markovian—potentially leading to exponential-time complexity in terms of the number of marginals—the proposed class can be transformed into a **Markovian** form. This transformation enables the use of efficient message-passing algorithms. The authors validated their methods through simulations and experiments on low-dimensional real-world problems.

## update after rebuttal

I would like to thank the authors for their detailed rebuttal. As stated previously, I find this paper interesting and continue to advocate for its acceptance.

**Claims And Evidence:**

### Unsupported Claims

Several claims in the paper lack sufficient justification:

1. **Motivation for Using Smooth Priors**
   The core motivation of the paper is to replace the standard Brownian motion-driven prior with a smooth one. While this idea seems reasonable, it is not thoroughly justified. Many trajectory inference problems indeed require smooth trajectories, but the authors should provide concrete examples illustrating when and why this is beneficial.

   Furthermore, the authors should explicitly state that their RNA-seq experiments are only toy versions, as they are reduced to dimensions 2 and 5 (a fact buried in the appendix). In reality, biological processes often exhibit strong microscopic randomness, making it unclear why smooth trajectories are preferable in this context.

2. **Continuity of Processes Relative to the Prior**
   A key missing question is: *What processes are absolutely continuous relative to the prior introduced in this paper?* This is crucial because, even if the prior is smooth, the minimizers of the Schrödinger bridge (SB) problem might still concentrate on non-smooth trajectories. If this is the case, it challenges the authors’ claim that they are performing *smooth* trajectory inference.

3. **Static vs. Dynamic Schrödinger Bridges**
   The authors appear to focus only on the **static** Schrödinger bridge, which corresponds to multi-marginal optimal transport with entropy regularization. While it is well known that the **values** of the static and dynamic SB problems coincide, recovering the latter from the former requires an additional flow-matching-style training step, which can be computationally expensive. This raises concerns about whether the comparisons in the experimental section are fair, especially regarding diffusion-based methods that train dynamic SBs directly.

4. **A Fundamental Limitation: Essentially a One-Dimensional Schrödinger Bridge**
   The most significant limitation of the proposed framework is that it **essentially reduces to a one-dimensional Schrödinger bridge**. The theory does not extend to high-dimensional Gaussian processes with dependent coordinates, which is a major drawback for real-world applications. This limitation likely explains why the experiments are restricted to problems of at most dimension 5.

**Essential References Not Discussed:**

The references are appropriately cited.

**Experimental Designs Or Analyses:**

The experimental design seems flawed; see my criticisms in **Claims And Evidence** and **Methods And Evaluation Criteria**.

**Methods And Evaluation Criteria:**

The experiments are primarily conducted on synthetic data and low-dimensional settings. The efficiency of the proposed algorithm in high-dimensional scenarios remains uncertain.

**Other Comments Or Suggestions:**

There is a "Table ??" at the bottom of p.19, line 1043.

**Other Strengths And Weaknesses:**

I have outlined several criticisms in **Claims and Evidence**. However, I believe this paper makes an interesting contribution due to the following strengths:

- **Novelty of the setting** – As the authors point out, introducing a smooth prior for the Schrödinger bridge problem is a novel and underexplored direction.
- **Simplicity of the algorithm** – The proposed algorithm is a straightforward modification of existing methods for Sinkhorn-type problems, making it both easy to understand and implement.
- **Clarity of presentation** – The paper is well-written and effectively communicates its ideas.

**Questions For Authors:**

Please see **Claims And Evidence**.

**Relation To Broader Scientific Literature:**

The key contribution of this paper is the introduction of a novel class of smooth priors for the Schrödinger bridge problem, which, to the best of my assessment, is both original and valuable.

**Theoretical Claims:**

The theoretical claims are sound and interesting.

---

> ### Author Rebuttal · Authors · 2025-03-31
>
> We thank the reviewer for these insightful comments.
>
> **Comment:** Which processes are absolutely continuous with respect to the GAP? Is the minimizer of the SB problem with GAP prior smooth?
>
> **Response:** When $R$ is a smooth prior, the solution to (1) is also smooth, in the same sense. To be more specific, a sample path of an order $m$ GAP is $m-1$-times differentiable with probabiliy 1. In other words, there is an event $A \subseteq \Omega$ such that each $\omega \in A$ lies in $C^{m-1}$ and $R(\Omega \setminus A) = 0$. Since the solution $P$ to (1) is necessarily absolutely continuous with respect to $R$, we must have that $P(\Omega \setminus A) = 0$ as well, so that $P$ is concentrated on $C^{m-1}$ paths. We will clarify this.
>
> Alternatively, the explicit form of the solution to the smooth SB problem given in Theorem 4.1 shows that the optimal $P$ is a mixture of Gaussian processes, which inherit the smoothness of $R$.
>
> **Comment**: Recovering the dynamic SB solution from the static one requires an additional flow-matching-style training step, which can be computationally expensive.
>
> **Response:** We would like to draw a distinction between two senses of "recovering" the dynamic SB, focusing for a moment on the standard SB. The *solutions* of the static and dyanmic SB problems actually coincide (by convolving the solution to the static problem with a Brownian bridge, see Leonard 2014). This is the argument we use in the proof of Lemma 2.1. In this sense, the dynamic SB is "recovered" directly from the static problem, and in particular, this allows easy sampling from the dynamic SB. However, the reviewer is correct that recovering the dynamic SB *as a stochastic process* (e.g., recovering the drifts in the corresponding SDE) is not so direct, though it can still be accomplished from the static problem (as in Pooladian & Niles-Weed, 2024).
>
> In our work, we recover a smooth dynamic SB in the first sense (with very reasonable computational complexity): given the solution to the static problem, we "recover" the corresponding dynamic solution in the sense that we can easily sample from it (using standard techniques in GP regression). In particular, we can sample from intermediate, unobserved times. (See our "leave-one-out" experiments in Tables 2 and 3, which shows that this leads to accurate reconstruction on synthetic data.)
>
> It is less clear how to recover it in the second sense, since the smooth dynamic SB is not given by the solution to an SDE. We agree that it would be interesting to pursue this direction in future work.
>
>
> **Comment:** What is the practical motivation for using smooth priors?
>
> **Response:** Our motivation is twofold:
>
> 1. In many systems, the only physically relevant models involve particles whose velocities vary continuously, i.e., whose paths are at least $C^1$. This is never the case with a BM prior. As our experiments show (e.g., Figure 4), this is particularly relevant when trajectories of two particles cross and separate: the BM prior easily loses the identity of particles between timesteps when two trajectories intersect, but assuming that the velocities are continuous allows the particles to be identified.
>
> 2. Existing work on trajectory inference largely focuses on smooth trajectories (see [F&S](https://proceedings.mlr.press/v130/chewi21a.html), [WLR](https://arxiv.org/abs/2405.19679) as examples), not for physical but for essentially *statistical* reasons. Like classic cubic splines, at a heuristic level, fitting data with smooth paths helps to ameliorate overfitting and ensure interpretability of the resulting estimates. Making this intuition rigorous in a particular statistical setting is a very interesting open problem.
>
> **Comment:**
> The most significant limitation of the proposed framework is that it essentially reduces to a one-dimensional Schrödinger bridge. The theory does not extend to high-dimensional Gaussian processes with dependent coordinates.
>
> **Response:**
> The fact that the theory does not extend to processes with dependent coordinates is not quite true.
> First, in Theorem 2.3, the covariance matrix $\sigma$ need not be diagonal; this allows the coordinates of the white-noise process defining the GAP to be correlated. More generally, if we consider a large covariance matrix $\mathbb{R}^{md \times md}$ that describes the covariance among all entries of hidden variables across dimensions, it is not necessary that this matrix be block diagonal (corresponding to independent coordinates). From an algorithmic perspective, this only changes the value of $\Gamma$ (see equations 19 and 20); the algorithm need not be changed.
>
> However, we agree that from a computational perspective, the use of independent coordinates is beneficial, since the tensor $\Gamma$ factors. This greatly reduces the space complexity of storing $\Gamma$, and simplifies some message passing updates.
>
> We also point the reviewer to our general comments above on dimension dependence.

---

### Official Review · Reviewer_N7xY · 2025-03-12

**Overall Recommendation:** 4

**Summary:**

The authors proposed a novel method to learn smooth trajectories in an SB problem, extending the usual SB method to allow non-Markovian reference by lifting it to phase space, that also effectively extending momentum SB. The method is accompanied with an approximated belief propagation algorithm.

**Claims And Evidence:**

The idea is well supported with theoretical results and empirical claim of algorithm is support by experimental evidences.

The only part being a bit hard to follow is that whether marginals are the continuous, population version or the discrete, sample version in theories. It might be helpful to spell that out more explicitly when using them.

**Essential References Not Discussed:**

Not I am aware of immediately.

**Experimental Designs Or Analyses:**

I checked all the experiment and did not find obvious issues.

**Methods And Evaluation Criteria:**

The evaluation criteria makes sense for empirical tests. The only potential complain I have is that the dimension tested are relatively low.

**Other Comments Or Suggestions:**

- There are missing cross references at then end of page 19 and caption of Table 16.
- I would like to have a full proof instead of proof sketch of Lemma 2.1, mostly to be sure about assumptions went in.

**Other Strengths And Weaknesses:**

Strengths
- The HMM view is very useful and I think it has potential to be extended to other setups beyond smoothness.
Weaknesses:
- I am unsure how scalable the algorithm to dimension is. Single cell data can have much higher dimensions. Even if not, having an algorithm work in several dimensions can be useful for a range of applications still.

**Questions For Authors:**

These questions might be out of the scope of the current paper but I am curious

1) it appears to me that the lifted SB problem having Markovian reference is key for the algorithm. It might be viewed as a missing data problem that we did not observe any higher order derivatives (which is momentum for momentum SB problem). Does the algorithm translate to general missing data problem that 1) some dimension are missing and 2) if they are observed the prior should be Markovian? Or is it also essential to have Gaussianity in GAP?

2) Is it true that an inferred trajectory necessarily pass one particle at each time point? This seems inherent from lemma 2.1. I understand this corresponding to interpolating empirical distributions. I wonder however, is this a desired property for all applications, e.g., in single cell examples/point cloud task one loses particle identity by killing the cell, so each (latent) trajectory is evaluated once and the chance a latent trajectory pass two observational data is zero, but in application e.g., individual particle tracking and N body problems one loses identity by permutation and we kept the same set of particles and we indeed want trajectories to pass exact one particle each snapshot. Is there an assumption that each time snapshots has same number of particles?

3) Related to 2), Lemma 2.1 (maybe also 3.1?) seems to assumed $\mu_k$ to be absolutely continuous, I wonder from when on the theories start to use empirical measures instead of population version?

**Relation To Broader Scientific Literature:**

The method extend momentum SB that are SoTA for smooth trajectory inference and can use quite general non-Markovian references. I think the latter is more important if it can be served as a foundation for more generalization to missing dimension problems.

**Theoretical Claims:**

I tried to check the proofs and did not find obvious errors. However Lemma 2.1 only has proof sketch and I want to see a fully spelled out proof for completeness. Especially spell out the assumptions on $\mu_k$, which appears to be continuous in the sketch but said to be general in lemma statement. In fact I would like the authors to double check if there is any missing assumptions for the marginals across the paper.

This is the main reason I put the current score and I am more than happy to change if assumptions are better spelled out.

---

> ### Author Rebuttal · Authors · 2025-03-31
>
> # Response to N7xY
>
> We thank the reviewer for these insightful comments.
>
> **Comment:**
> What is the assumptions on the marginals in Lemma 2.1 (and 3.1)? In fact I would like the authors to double check if there is any missing assumptions for the marginals across the paper.
>
> **Response:**
> Thank you for allowing us to clarify this important point, which we'll make clearer in the revision.
>
> A complete proof of Lemma 2.1 is now included in the appendix. Lemma 2.1 consists of two parts:
>
> 1. The first part demonstrates that the multi-marginal Schrödinger bridge problem with finite marginal constraints (1) can be reduced to a finite-dimensional KL divergence minimization problem (2). This reduction requires no assumptions about the marginals.
>
> 2. The second part addresses a technical challenge: we want to define a multi-marginal Schrödinger bridge problem with the prior $R$ being a GAP and the marginals being discrete. However, the KL divergence in the optimization problem is ill-defined when $R$ has absolutely continuous marginals and the marginals are discrete. We address this by proving that problem (2) is equivalent to formulation (3) *if* the marginals have densities. Since (3) is well defined even for discrete marginals, we take (3) as the **definition** of the multi-marginal Schrodinger Bridge for general marginal distributions. This is in fact a standard approach in statistical analysis of the Schrodinger Bridge problem, see, e.g., Pooladian & Niles-Weed (2024).
>
> The equivalence between (2) and (3) uses two assumptions:
> * The marginals $\mu_k$ are absolutely continuous with respect to the Lebesgue measure, with finite entropy $\int \mu_k(x) \log \mu_k(x) dx$
> * The joint marginals of the prior $R$ are absolutely continuous
>
> In the remainder of the paper, we adopt the setting described in the beginning of Section 2.2 where the marginals are sampled data (i.e., $\mu_k$ is discrete). In particular, this is true of Lemma 3.1. Note that, before the lemma, we defined $p(x, y)$ to refer to a "mixed" discrete-continuous density, where $x$ coordinates are discrete but $y$ coordinates are continuous. We will emphasize this point in the revision.
>
> **Comment:**
> Is the Markovian assumption in the GAP prior essential? How about the Gaussianity?
>
> **Response:**
> The GAP itself is not Markov; however, the Markovian structure of the **lifted** process is crucial for our algorithm because:
>
> 1. It enables factorization of the joint distribution into pairwise terms, allowing efficient message passing.
>
> 2. It ensures the solution $p(z)$ of (6) is Markovian (by Theorem 4.1), enabling efficient trajectory sampling.
>
> Gaussianity in the GAP prior provides computational efficiency through closed-form message passing updates but isn't essential. The method can be generalized to work with any process that can be lifted to a Markov process on the phase space by replacing the potential functions between $\eta_k$ and $\eta_{k+1}$ with the conditional density $r(\eta_{k+1} | \eta_k)$ of the desired prior process.
>
> **Comment:**
> Does the algorithm translate to general missing data problem?
>
> **Response:**
> **Yes**. As the reviewer mentioned, not observing higher-order derivatives is a form of missing data handled by our algorithm. Other forms of missing data can be incorporated by modifying the factor nodes $\alpha_k$ in the graphical model (Figure 3). Currently, these nodes are indicators enforcing that only positions $\omega_k$ are observed. By modifying these factor nodes, one can incorporate other types of missingness in the observations. We plan to pursue this in future work.
>
> **Comment:**
> Is it true that an inferred trajectory necessarily pass one particle at each time point? And does is makes sense in the context of single cell example? Is there an assumption that each time snapshots has same number of particles?
>
> **Response:**
> Yes, in the current formulation, inferred trajectories must pass through exactly one particle at each observed time point, due to the marginal constraints.
>
> For point cloud matching applications, while tracking individual "particles" may not be physically meaningful, our algorithm remains valuable for inferring collective motion patterns:
>
> 1. We can infer velocity and acceleration fields that characterize point cloud dynamics
> 2. We can interpolate the point cloud distribution at unobserved times by sampling from the solution
>
> Our results in Table 2 show that this approach yields good results.
>
> Relaxing the requirement that trajectories pass through observed points can be achieved by modifying the factor nodes $\alpha_k$ to model observational noise. This allows inferred trajectories to deviate from exact observed positions. We've added this extension as a remark in the revision, and plan to pursue it in follow-up work.
>
> Our method does not require the same number of particles across different time snapshots. Both the theoretical framework and algorithm support arbitrary discrete marginal distributions.

---

> > ### Comment · Reviewer_N7xY · 2025-04-01
> >
> > I really appreciate the authors' response! I have raised my score.
> > - Especially the clarification on discrete vs continuous marginals. I asked this mainly because estimating the bridge between two continuous marginals while only having samples from them is subtly different from estimating the bridge between the two corresponding empirical measures e.g. to get the first you do not have to have trajectories pass at least one particle each marginal and some of the half-bridge based methods do not require that. Please make sure to have them clarified in the revision on which assumptions are used in each result.
> >
> > - Thank you for the answer to the missing data generalization. The reason I raised this question to start with is that I was not very convinced that in scRNA-seq applications we want smoothness given the intrinsic noise in scRNA-seq data is from single-molecule chemical reactions and is inherent. However, missing data is common, e.g. scRNA-seq cannot measure protein level but in reality, protein is the one that actually regulates RNA. E.g. a model widely accepted by biologists in [1] (section *BoolODE: converting Boolean models to ODEs.*) has protein components in it that are not observable. The only work I know tried to deal with this is [2] (app. C.4) but used a quite different idea, parametric models and the goal is not quite trajectory inference. So I am glad the authors' work could lead to a solution. I look forward to the authors' follow-up work :). But please make some comments on this potential as it might attract biologists.
> >
> > Again, thank the authors for this nice piece of work, well done.
> >
> > [1] Pratapa, Aditya, Amogh P. Jalihal, Jeffrey N. Law, Aditya Bharadwaj, and T. M. Murali. "Benchmarking algorithms for gene regulatory network inference from single-cell transcriptomic data." Nature methods 17, no. 2 (2020): 147-154.
> >
> > [2] Berlinghieri, Renato, Yunyi Shen, and Tamara Broderick. "Beyond Schrödinger Bridges: A Least-Squares Approach for Learning Stochastic Dynamics with Unknown Volatility." In 7th Symposium on Advances in Approximate Bayesian Inference -- Workshop Track.

---

> > > ### Author Response · Authors · 2025-04-02
> > >
> > > Thank you very much for the helpful references---this seems like a quite promising direction! We definitely plan to comment on this in the revision, and are very grateful for the pointers.

---

### Official Review · Reviewer_XtBd · 2025-03-13

**Overall Recommendation:** 3

**Summary:**

The paper proposes solving multi-marginal Schrödinger bridges w.r.t. a reference process based on autoregressive Gaussian process. Interpretation to phase space is constructed, which leads to a tractable algorithm based on probabilistic graphical models and belief propagation.

**Claims And Evidence:**

- In the end of Sec 2.2 and Footnote 1, it's a bit unclear why "smooth" paths corresponds to stationary processes. As mentioned by the authors before Sec 3, there are a few prior works that encourage smoothness through non-stationary processes.

Overall, I think this paper presents an interesting theoretical finding yet falls short on the experiment sides.

**Essential References Not Discussed:**

N

**Experimental Designs Or Analyses:**

- Experiment results are in rather low dimension (<= 5). Dimension reduction is performed on some sc-RNAseq datasets, e.g., Embryoid Body was initially reported in MIOFlow with dimension 200. How does the complexity of the proposed method scales in terms of dimension?

**Methods And Evaluation Criteria:**

Y

**Other Comments Or Suggestions:**

Typo in L268: Sec 3 should be Fig 3

**Other Strengths And Weaknesses:**

N/A

**Questions For Authors:**

Can the authors comment on the relation to https://arxiv.org/pdf/2006.14113? This prior work also introduced a belief-propagation based method for solving multi-marginal OT.

**Relation To Broader Scientific Literature:**

Numerical methods for solving trajectory inference problems could be beneficial, as these problems are widespread across various scientific domains.

**Theoretical Claims:**

Y

---

> ### Author Rebuttal · Authors · 2025-03-31
>
> # General remarks
>
> We thank the reviewers for their helpful comments and questions. We are gratified that the reviewers found our method "interesting" (XtBd), "novel," "very useful" (N7xY), and theoretically "sound" (okGt).
>
> We would like to address an important issue that arose in several reviews: **dimension dependence**.
>
> First, we agree that we should highlight more explicitly that our experiments were conducted on low-dimensional data. We plan to include this fact prominently in our revision.
>
> Second, the reviewers are correct that the dependence on dimension is poor. The running time scales quadratically with $M$, the number of coefficients used in the approximation algorithm (see Theorem 6.1). As mentioned in Section 6, $M$ will typically be exponential in the dimension, limiting our approach to relatively low dimensional problems.
>
> However, we offer several arguments supporting the interest of our method:
>
> 1. A low-dimensional algorithm is still valuable in applications. For example, in the fluid dynamics and astronomical object tracking examples mentioned in the introduction, the data lies in dimension 2 or 3. Our method makes an important contribution to these inherently low-dimensional problems. Based on our experiments, smooth SBs are now the leading method for N-body tracking problems in 3 dimensions.
>
> 2. Existing approaches often fail, even in low dimension. In the 5-dimensional Dyngen dataset, our algorithm (Figure 5) provides significantly more reasonable results than current leading approaches (F&S, Figure 7; MIO, Figure 8). Another algorithm, DMSB, fails to even converge (Table 2). This indicates the low-dimensional setting is far from solved.
>
> 3. A small number of coefficients go a long way. In our experiments on the 5-dimensional Dyngen dataset, we use 1024 coefficients (only 4 per dimension), yet our approach still brings substantial benefits. We have also conducted experiments on 10-dimensional data using 1024 coefficients. Our method outperforms the other methods on this data in a leave-one-out task. We will include this new, higher-dimensional experiment in our revision.
>
> 4. It is common to assume that high dimensional data has low intrinsic dimension. Many approaches to trajectory inference begin with non-linear dimension reduction. For example, the pipline developed by Schiebinger et al. (2019) begins by reducing dimension to 30. While this is still larger than we can handle, "working up" from low-dimensional examples can yield biologically relevant insights as computational power improves.
>
> # Response to XtBd
>
> **Comment:**
> In the end of Sec 2.2 and Footnote 1, it's a bit unclear why "smooth" paths corresponds to stationary processes.
>
> **Response**
> The footnote about stationary processes has been removed. To clarify, we've added a new lemma in the appendix proving that any Markov Gaussian process with differentiable paths is essentially trivial:
>
> Let $\omega: t \mapsto \omega(t)$ be a real-valued Gaussian process that is Markovian and a.s. differentiable. For arbitrary $t \ge 0$ such that $\mathrm{Var}(\omega(t)) > 0$, we have $\omega(s) = \mathbb{E}[\omega(s)|\omega(t)]$ for all $s \ge 0$ a.s.
>
> In other words, any such Gaussian process is essentially deterministic, indicating that obtaining smooth paths requires moving beyond Markov processes.
>
> **Comment:**
> Relation to [2006.14113]?
>
> **Response:**
> We appreciate the reviewer mentioning this important work, which we failed to cite though we cited its companion works (''Learning hidden markov models from aggregate observations'' and ''Multimarginal optimal transport with a tree-structured cost and the Schrödinger bridge problem"), which have the same authors.
>
> In [2006.14113], the authors develop a belief propagation algorithm for multi-marginal optimal transport. The use of belief propagation for multi-marginal OT has a long history (Teh & Welling, 2001), and like [2006.14113], we use this connection for an efficient algorithm. However, we believe we're the first to exploit this connection to develop an algorithm for the Schrodinger Bridge problem with a smooth prior.
>
> Our Algorithm 2 is indeed analogous to Algorithm 2 in [2006.14113], tailored to our setting:
> 1. The potential functions are:
>    - Between $\eta_{k-1}$ and $\eta_k$: Given by $\Phi_k$ (defined in Section 4)
>    - Between $\eta_k$ and $\omega_k$: Given by the indicator function $\mathbf{1}_{\{z_k^{(0)} = x_k\}}$
> 2. Our message passing variables ($\delta^-$, $\delta^+$, $\beta$, $\gamma$) correspond to their factor-to-variable messages $m_{\alpha \rightarrow j}
> 3. Our operators $\mathcal{L}$, $\mathcal{I}$, $\mathcal{R}$ implement the operations in equations (38a,b)
>
> An important technical difference is that since our $\eta_k$ variables are continuous, the operators $\mathcal{L}$, $\mathcal{I}$, $\mathcal{R}$ cannot be directly implemented. This fact requires us to develop an approximation scheme, which gives rise to our practical implementation detailed in Section 5.

---

### Decision · Program_Chairs · 2025-05-01

**Decision:**

Accept (poster)

**Comment:**

This paper considers trajectory inference in the space of probability measures. The traditional approach of multi-marginal Schrodinger bridge is improved by the introduction of a class of smooth Gaussian processes as priors. Efficient algorithms are developed based on lifting and, consequently, (approximate) belief propagation. Reviewers and AC largely found the idea interesting, the method effective, and the work rigorous, although multiplier reviewers also had concerns about scalability with dimension. Overall, the merits overweigh imperfection, and I recommend acceptance as poster (not higher because the concerns around scalability have not been completely resolved).